# Molecular interplay of ASNS and the PI3K-AKT-mTOR pathway in CMV and HIV co-infections: Therapeutic implications

Hao Zhang[1�], ShuYou Yuan[2�], ShaoXiang Ding[1], HongXia Bao[1], WenJun Chen[3], Bo Cai[4], JunKai Sun[5*], HaoGang Zhu[1*], Wei Lu[1*], Ye Fang[6*]

1 Geriatrics Center, Wuxi Second Geriatric Hospital, Wuxi, Jiangsu, China, 2 Laboratory Department, Wuxi Second Geriatric Hospital, Wuxi, Jiangsu, China, 3 Neurology Department, Wuxi Second Geriatric Hospital, Wuxi, Jiangsu, China, 4 Pathology Department, Wuxi Second Geriatric Hospital, Wuxi, Jiangsu, China, 5 Department of Interventional Radiology, Wuxi No. 5 People's Hospital, Wuxi, Jiangsu, China, 6 Department of Emergency Medicine, Wuxi People's Hospital, Wuxi, Jiangsu, China

☯ These authors contributed equally to this work.
* fangye841014@sina.com (YF); luwei5166@gmail.com (WL); 911207258@qq.com (HZ); 854437824@qq.com (JS)

## Abstract

CMV/HIV coinfection markedly exacerbates disease progression, elevates treatment failure risk, and worsens patient outcomes, yet the underlying molecular mechanisms remain incompletely understood—creating an urgent need for targeted host-focused research. This study identifies asparagine synthetase (ASNS) as a pivotal metabolic-signaling hub in coinfection pathogenesis, with critical interactions with the PI3K-AKT-mTOR pathway. Using integrated bioinformatics analyses of transcriptomic data, ASNS emerged as a central hub in protein-protein interaction networks, with robust positive co-expression alongside key PI3K-AKT-mTOR components (PIK3CA, MTOR, AKT2, AKT3), while machine learning validated AKT2 as a critical node. ASNS was consistently upregulated 48 hours following CMV infection and across all HIV disease stages, while single-cell RNA sequencing localized ASNS and MDM2 to plasma cells in HIV-positive individuals—implicating their role in virus-driven immune responses. Transcription factor analysis identified RUNX1 as a central regulator: bioinformatics predictions confirmed RUNX1 binds to the ASNS promoter, and validation studies identified RUNX1 as the top biomarker for HIV treatment resistance (AUC = 0.714). Molecular docking and 200-ns dynamics simulations showed that cidofovir—an approved antiviral agent—binds ASNS with high affinity (−6.61 kcal/mol) through nine hydrogen bonds, forming a more stable complex than ASNS-ONL, with VAL-51 and ASN-74 as key residues. Collectively, these findings establish ASNS as a host metabolic-signaling hub exploited by CMV and HIV, highlighting its potential as a novel therapeutic target. Targeting ASNS, particularly at residues VAL-51 and ASN-74, may offer a promising host-directed strategy to improve coinfection

**Data availability statement:** All relevant data and R code for machine learning analyses are available within the paper, Supporting Information files, or upon request from the corresponding author via email.

**Funding:** This work was supported by four grants awarded to Hao Zhang: the Wuxi Science and Technology Development Fund (Award Number: Y20232005); the 2025 Open Project of the Provincial Key Laboratory for Integrative Chinese-Western Prevention and Treatment of Geriatric Diseases at Yangzhou University (Award Number: 202528); the 2025 Wuxi Aging Research Project (Award Number: WXLN25-A-24); and the 2025 Annual Scientific Research Project of the Jiangsu Provincial Association of Geriatrics (Award Number: JGS2025ZDM010). These fundings enabled the comprehensive investigation of the molecular mechanisms underlying ASNS in CMV and HIV co-infections and facilitated the development of potential therapeutic strategies.

**Competing interests:** The authors have declared that no competing interests exist.

treatment outcomes. This work lays the groundwork for experimental validation and the development of targeted therapies for CMV/HIV coinfection.

## Introduction

Cytomegalovirus (CMV) and Human Immunodeficiency Virus (HIV) coinfection represents a major clinical burden, with CMV seroprevalence among HIV-positive individuals reaching up to 90% [1]. This high rate remains prevalent owing to HIV-induced immunosuppression, which impairs immune surveillance and fuels CMV reactivation—creating a synergistic pathogenic cycle: reactivated CMV further exacerbates immune dysfunction, leading to severe complications such as retinitis, encephalitis, and gastrointestinal disorders that significantly compromise patient quality of life and survival [2]. Despite this profound clinical impact, the molecular underpinnings of CMV/HIV coinfection remain incompletely understood [3].

Most existing studies focus on CMV or HIV in isolation, relying on single-virus datasets that fail to capture the complex crosstalk and shared pathogenic factors between the two infections [4]. While both viruses are known to hijack host metabolic and signaling pathways to support their replication and evade immune surveillance, the shared molecular mechanisms driving their synergistic pathogenesis during coinfection remain undefined [5]. This critical knowledge gap hinders the development of effective therapies for coinfected patients, who currently depend on treatments plagued by limitations—including ganciclovir resistance in CMV and adverse side effects associated with long-term HIV treatment regimens [6].

Asparagine synthetase (ASNS), a key enzyme in amino acid metabolism, plays well-established roles in the cellular response to viral infections [7]. It regulates protein synthesis—an essential process for viral replication—and modulates immune evasion, both of which are central to the pathogenesis of both CMV and HIV [8]. Notably, ASNS interacts closely with the PI3K-AKT-mTOR pathway, a central signaling hub that both viruses hijack to sustain their survival and replication [9]. As a core hub in protein-protein interaction networks linked to both infections, ASNS emerges as a potential critical mediator of the shared pathogenic mechanisms in CMV/HIV coinfection—yet its specific role in this context remains unexplored [10].

Here, we hypothesize that ASNS functions as a central metabolic-signaling hub exploited by both CMV and HIV during coinfection, via its interaction with the PI3K-AKT-mTOR pathway. The primary objective of this study is to elucidate the molecular mechanisms by which ASNS contributes to CMV/HIV coinfection, identify its regulatory networks (including transcription factors such as RUNX1), and validate its potential as a therapeutic target.

To achieve this goal, we used an integrated bioinformatics approach leveraging publicly available gene expression datasets, and detailed information for all GEO datasets employed in our analyses is provided in **Table 1**. This approach integrates differential gene expression analysis, protein-protein interaction (PPI) network construction, and machine learning to facilitate unbiased identification of shared

**Table 1. Characteristics of GEO datasets utilized for investigating the common pathological mechanisms of CMV and HIV infections.**

| Dataset (Year) | Study Focus | Sample (Virus/Cell/N) | Platform (Tech) | Major Findings |
|---|---|---|---|---|
| GSE14490 (2009) | CMV-infected dendritic cells | CMV/ MoDCs/ N = 20 | Agilent (Microarray) | CMV-induced immunosuppression via DC modulation |
| GSE68563 (2016) | HIV-1/2 infected PBMCs | HIV-1/2/ PBMCs/ N = 9 | Agilent (Microarray) | Differential regulation of host factors by HIV-1 and HIV-2 |
| GSE81246 (2017) | Symptomatic CMV infection | CMV/ PBMCs/ N = 48 | Affymetrix (Microarray) | Massive NK/T cell expansion in symptomatic infection |
| GSE6740 (2007) | HIV progression T cell atlas | HIV/ T cells/ N = 20 | Custom (Microarray) | Persistent IFN response and thymic output signature |
| GSE33580 (2012) | HIV-exposed uninfected women | HIV/ Blood/ N = 86 | Affymetrix (Microarray) | HIV resistance linked to glycolysis downregulation |
| GSE157829 (2020) | Immune exhaustion in HIV | HIV/ PBMCs/ N = 7 | Illumina (scRNA-seq) | Identification of KLRG1 + exhausted CD8 + T cells |

Abbreviations: CMV, Cytomegalovirus; MoDCs, Monocyte-derived Dendritic Cells; PBMCs, Peripheral Blood Mononuclear Cells; scRNA-seq, single-cell RNA sequencing; IFN, Interferon.

molecular nodes across CMV and HIV infections, overcoming the limitations of single-virus experimental models. Our research workflow is depicted in **Fig 1**.

## Materials and methods

### Bioinformatics analysis of differential gene expression and functional enrichment for CMV and HIV infection

To identify host factors associated with CMV and HIV coinfection, this study integrated transcriptomic differential analysis, protein-protein interaction (PPI) network construction, machine learning-based screening, and functional enrichment analysis using the GSE14490 and GSE68563 datasets. GSE14490 (CMV-infected immature monocyte-derived dendritic cells) had differentially expressed genes (DEGs) identified via the limma package in R 4.2.1 with stringent criteria ($|logFC| > 0.58$, p.adj < 0.05), while GSE68563 (HIV-1-infected peripheral blood mononuclear cells) used a p < 0.05 threshold for DEG selection. Data quality control excluded probes mapping to multiple molecules and retained the highest-signal probe for molecules with multiple probes, with DEGs from both datasets visualized via volcano plots to summarize expression changes. Venn diagram analysis (VennDiagram package, R 4.2.1) identified shared/unique DEGs, revealing 42 overlapping genes used to construct PPI networks via STRING (v11.0, high-confidence threshold = 0.7 for reliable interactions) and visualized/analyzed in Cytoscape (v3.7.2). Key hub genes in PPI networks were identified via the CytoHubba plugin using degree, maximum clique centrality (MCC), and betweenness centrality algorithms—selected for prioritizing highly connected, functionally critical nodes. Random forest analysis (randomForest package, R 4.2.1; 100 decision trees) evaluated gene importance in the critical PPI module, chosen for robustness in prioritizing infection-linked genes from high-dimensional transcriptomic data. Gene Ontology (GO) and Kyoto Encyclopedia of Genes and Genomes (KEGG) enrichment analyses used clusterProfiler (v4.4.4) and GOplot (v1.0.2) in R; Gene Set Enrichment Analysis (GSEA) for the CMV dataset (via clusterProfiler) explored associated biological pathways.

### Analysis of ASNS and the PI3K-AKT-mTOR pathway in CMV infection

To investigate ASNS's role, its interactions with the PI3K-AKT-mTOR pathway, and immune checkpoint gene expression profiles during CMV infection, we analyzed time-series data from the GSE14490 dataset—non-infected and 6h/48h post-infection immature monocyte-derived dendritic cells from CMV-negative donors—focusing on non-infected and 48h samples for core analyses. Recognizing the dataset's small sample size as a limitation, we used leave-one-out

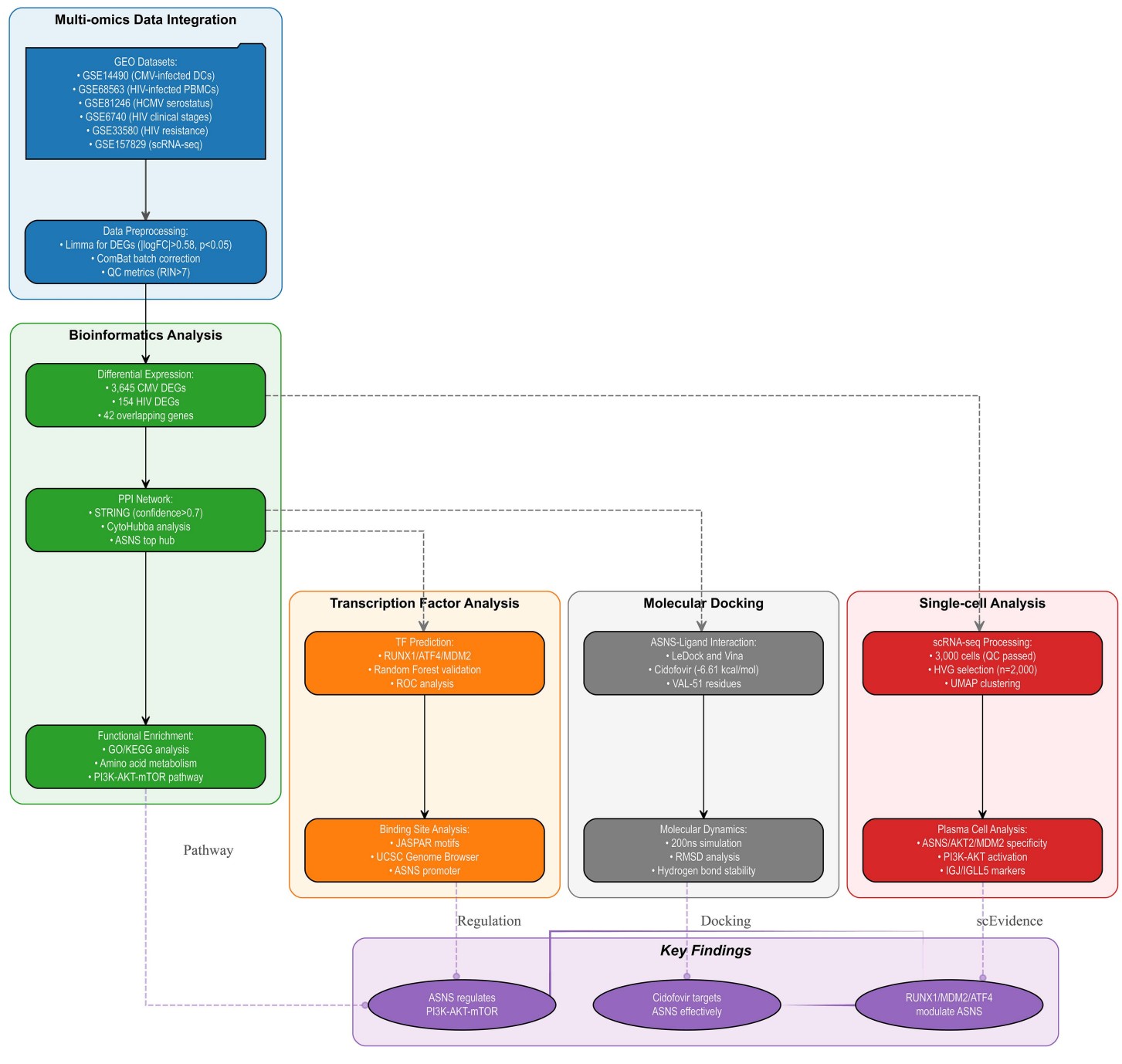

**Fig 1. Workflow of the dtudy.**

cross-validation (LOOCV) for model training/evaluation, a method optimal for small cohorts. For co-expression analysis, we selected ASNS and key PI3K-AKT-mTOR molecules (PIK3CA, MTOR, AKT1–3) [11]—chosen for the pathway's established viral infection regulatory role—with Pearson/Spearman correlations (significance via p-values) and heatmaps generated using ggplot2 (v3.4.4) in R (v4.2.1). Differential expression and immune checkpoint analyses (comparing

non-infected/48h samples to characterize functional status) were performed in R (v4.2.1 for differential analysis, v4.0.3 for immune checkpoint analysis) using limma and a p < 0.05 threshold. Three machine learning models (random forest, SVM, XGB) were trained via R's caret, randomForest, xgboost, and e1071 packages to identify PI3K-AKT-mTOR molecules associated with ASNS, with performance assessed using RMSE, R-squared, and MAE—metrics robust for small-sample high-dimensional data.

## Multi-dataset analysis of ASNS and the PI3K-AKT-mTOR pathway in CMV and HIV infections

To investigate ASNS's role and associations with the PI3K-AKT-mTOR pathway (PIK3CA, MTOR, AKT1–3) in CMV and HIV infections, we analyzed three gene expression datasets: GSE81246 (CMV-focused: PBMCs from 12 seronegative, 25 seropositive healthy donors, 11 symptomatic primary CMV-infected adults), GSE6740 (HIV-focused: CD4+/CD8+T cell transcriptomes from untreated HIV-infected individuals, non-progressive cases with low/undetectable viral loads, and healthy controls), and GSE33580 (HIV treatment-resistant individuals with high viral loads/virologic failure). All datasets underwent preprocessing (normalization, z-score transformation, missing value handling, sample ID standardization). Spearman correlation analysis (robust for non-normal gene expression data) assessed ASNS-pathway molecule relationships, with co-expression heatmaps generated via ggplot2 in R 4.2.1. For HIV-specific analyses using GSE6740, leave-one-out cross-validation (LOOCV, optimal for small cohorts) was used for model training/evaluation (acknowledging small sample size as a limitation). Three machine learning models (random forest, SVM, XGBoost) were trained via R's caret, randomForest, and xgboost packages to identify pathway molecules linked to ASNS, with performance assessed via RMSE, R-squared, and MAE (reliable for high-dimensional small-sample data).

## Single-cell RNA sequencing analysis of HIV infection

Single-cell RNA sequencing (scRNA-Seq) data from the GEO database (accession: GSE157829) were preprocessed and curated using the Seurat package (v4.1.0) in R. The dataset included 1 HIV-negative reference sample (GSM4775588) and 5 HIV-infected experimental samples (GSM4775591, GSM4775593, GSM4775589, GSM4775592, GSM4775594). Stringent quality control filtered out genes detected in <3 cells or <100 cells (post-initial filtering) and cells with <200/>2500 detected genes or mitochondrial content >10%, yielding 3,000 high-quality cells for analysis (acknowledging small cell count as a study limitation). UMI counts were normalized with a 10,000 scaling factor, log-transformed, and scaled using Seurat's ScaleData function. Post-preprocessing, functional analysis explored HIV-induced cellular landscape alterations: batch effects were corrected via harmonization, 2,000 highly variable genes (HVGs) were selected based on standard deviation, and normalized data underwent principal component analysis (PCA) focusing on the top 10 variable genes. Uniform Manifold Approximation and Projection (UMAP) and clustering (resolution = 0.3, Seurat) utilized the top 15 principal components for visualization and cellular population identification, with clusters annotated via the top 10 marker genes per cluster. Pathway score comparisons between cell groups were computed using Seurat's AverageExpression and the GSVA package, with results visualized via pheatmap.

## Comprehensive analysis of transcription factors regulating ASNS in CMV and HIV infections

HIV infection directly stimulates plasmacytosis and hypergammaglobulinemia [12]. In contrast, cytomegalovirus indirectly modulates humoral immunity through targeting plasmacytoid dendritic cells [13]. These pathogen-driven responses are supported by intracellular amino acid metabolism, notably glutaminolysis, which is essential for plasma cell longevity and antibody synthesis [14]. To identify transcription factors regulating ASNS and contributing to plasma cell differentiation in HIV patients, we analyzed four key genes (IGJ, IGLL5, MZB1, ASNS) via integrated bioinformatics approaches. Potential transcription factors were predicted using the Chipbase database, followed by random forest (RF) analysis—selected for robust prioritization of regulatory factors in transcriptomic data—on the GSE14490 (CMV-infected immature monocyte-derived dendritic cells: non-infected/48h post-infection) and GSE68563 (HIV-1-infected PBMCs) datasets. The

RF model (100 decision trees, consistent seed for reproducibility) used Mean Decrease Gini to assess transcription factor significance, with ROC analysis (pROC package, R 4.2.1) on GSE14490 evaluating predictive accuracy. RUNX1-ASNS molecular interactions were explored via JASPAR (binding site prediction) and UCSC Genome Browser (ASNS promoter region mapping) to infer direct promoter binding. Additional analyses included GSE81246 (HCMV seronegative/sero-positive donors, symptomatic primary HCMV infection), GSE33580 (HIV-resistant individuals with virologic failure), and GSE157829 scRNA-seq data. ScRNA-seq preprocessing (Seurat, R) involved low-quality cell filtering and normalization (3,000 high-quality cells retained, small cell count acknowledged as a limitation), with UMAP for dimensionality reduction/visualization to characterize MDM2 expression across cell types and differential expression analysis comparing MDM2 levels between HIV-positive and HIV-negative individuals.

## Microarray processing, feature analysis, and biomarker diagnostic utility assessment for HIV infection

Raw microarray data from GSE33580 (Affymetrix HG-U133 Plus 2.0 Array; 43 HIV-resistant, 43 HIV-negative samples) were processed in R v4.2.1 using the affy package: CEL files imported via ReadAffy(), followed by background correction, quantile normalization, and log2 transformation with the RMA algorithm. Quality control (QC) included missing value percentage, zero-variance gene proportion, median sample coefficient of variation (CV), pairwise Pearson correlations, and expression range, with standardized distributions visualized via ggplot2. For five target features (RUNX1, MDM2, ATF4, ASNS, AKT2)—linked to transcription factor regulation in HIV-related contexts—preprocessing involved median imputation of missing values and factor conversion of group labels. Feature importance was computed across random forest (importance metric), logistic regression (glmnet coefficient magnitude), and XGBoost (xgb.importance())—results averaged and normalized to 0–1. Diagnostic performance of individual biomarkers was evaluated via ROC curves and AUC (pROC::roc()). Clinical utility was assessed through decision curve analysis (DCA) and clinical impact curve (CIC) using rmda::decision_curve() and rmda::plot_clinical_impact() (threshold range: 0.0–1.0). A clinical prediction nomogram was constructed via rms::nomogram() based on logistic regression, with performance quantified by AUC.

## PCA and machine learning model performance assessment for HIV infection datasets

Data from the GSE33580 dataset (43 HIV-resistant, 43 HIV-negative samples) were processed in R via median imputation (selected for handling random missing values in gene expression data) and Z-score standardization (to eliminate inter-sample dimensional differences). Principal component analysis (PCA) was performed using FactoMineR::PCA() to assess sample clustering and feature contribution, with visualization via factoextra::fviz_pca_biplot(). Nine machine learning algorithms (including LDA, Decision Tree, XGBoost, Naive Bayes) were implemented through the caret package, with 5-fold repeated cross-validation (trainControl(method = "repeatedcv", number = 5, repeats = 2))—a strategy balancing bias and variance to enhance result reliability. Model performance was evaluated using AUC, accuracy, sensitivity, specificity, and F1 score, calculated via pROC::roc() and caret::confusionMatrix().

## Molecular docking analysis of small molecule-ASNS protein interactions using LeDock and AutoDock Vina

To explore small molecule binding potential with the ASNS protein (PDB ID: 6gq3)—a key mediator of CMV/HIV coinfection via the PI3K-AKT-mTOR pathway—we performed molecular docking using two complementary programs (LeDock, AutoDock Vina) to enhance result reliability. Three small molecules were selected: ONL (5-OXO-L-NORLEUCINE, extracted from ASNS's PDB structure 6gq3), Bisabosqual A (PubChem CID: 9821444, a known ASNS interactor) [15], and Cidofovir (PubChem CID: 60613, predicted via the Comparative Toxicogenomics Database [CTD]) [16]. For LeDock, docking protocol validity was confirmed via ONL redocking: Root Mean Square Deviation (RMSD) between docked and original ligand poses was calculated using PyMOL's align function, verifying accurate reproduction of native binding conformations. Bisabosqual A and Cidofovir were subsequently docked, with RMSD values (vs. original ONL pose) computed via PyMOL; 2D diagrams of optimal binding conformations were generated using Discovery Studio. For AutoDock Vina,

ligand preparation (Discovery Studio) included hydrogen addition, Gasteiger charge assignment, and energy minimization to ensure conformational suitability. The docking site was defined by XYZ coordinates (24.5813, 16.1878, 42.9525 Å) following ONL redocking via the Vina plugin. Docking utilized Vina's scoring function to predict binding affinity, with the grid box centered on the predefined active site; results were analyzed and protein-ligand interactions visualized using PyMOL. All docking experiments were performed in triplicate to ensure reproducibility.

## Molecular dynamics simulations of ASNS-ONL and ASNS-Cidofovir complexes

To validate the stability and binding interactions of ASNS (PDB ID: 6gq3) with ONL and Cidofovir, molecular dynamics (MD) simulations were performed using GROMACS 2022. Complex structures were derived from AutoDock Vina docking results, with two systems analyzed: ASNS-ONL and ASNS-Cidofovir. The GAFF force field was selected for ligand parameterization (optimized for small molecule flexibility) and AMBER14SB for the protein (widely validated for globular protein simulations), paired with the TIP3P water model—a combination ensuring accurate representation of biomolecular interactions. Simulation systems were prepared via energy minimization, followed by 100 ps of NVT equilibration (V-rescale algorithm, 298 K) and 100 ps of NPT equilibration (Berendsen barostat, 1 bar) to stabilize temperature and pressure. Production MD runs were conducted for 200 ns with a 2 fs time step, saving conformations every 10 ps. Electrostatic interactions were computed using the Particle-Mesh Ewald (PME) method (1.2 nm cutoff), and non-bonded interactions were truncated at 1.0 nm (updated every 10 steps)—parameters balancing computational efficiency and interaction accuracy. Post-simulation analyses (binding stability, interaction dynamics) and visualization were performed using VMD and PyMOL.

## Results

### Differential gene expression and functional enrichment analysis of ASNS in CMV and HIV infections

Volcano plot analysis of the CMV dataset (GSE14490) identified 3,645 differentially expressed genes meeting |logFC|>0.58 and p.adj<0.05 (Fig 2A), with ASNS, AKT2, and AKT3 significantly upregulated, while 154 differentially expressed genes were detected in the HIV dataset (GSE68563) (Fig 2B), including notable ASNS upregulation—consistent ASNS elevation across both viral infections points to a correlative role in shared pathogenic signatures. Venn diagram analysis revealed 42 overlapping genes between the two datasets, indicating a conserved transcriptional response to CMV and HIV infection (Fig 2C). PPI network analysis of these overlapping genes using CytoHubba's degree, MCC, and betweenness centrality algorithms all ranked ASNS as the top hub gene (Fig 2D–2F). Random forest (RF) analysis of the CMV dataset further confirmed ASNS as the most influential gene in the critical gene module, both at the 48-hour post-infection time point and in the combined analysis of non-infected, 6-hour, and 48-hour post-infection samples, supported by high Mean Decrease Gini scores (Fig 2G–2H). Enrichment analyses linked ASNS to amino acid metabolism pathways: in the CMV dataset, ASNS was associated with the "biosynthesis of amino acids" pathway (hsa01230) (Fig 2I) and significantly enriched in the "Metabolism of Amino Acids and Derivatives" pathway via GSEA (normalized enrichment score = 1.493, p = 0.043; Fig 2J); in the HIV dataset, ASNS similarly mapped to hsa01230 (Fig 2K); and enrichment analysis of the 42 overlapping genes highlighted ASNS's central role in the shared "Amino Acid Metabolism" pathway (Fig 2L). Collectively, these findings from multiple analytical approaches across CMV and HIV datasets demonstrate consistent ASNS upregulation, its status as a core hub gene, and its correlation with amino acid metabolism, underscoring conserved molecular signatures relevant to both viral infections.

### Expression analysis of ASNS and key molecules in the PI3K-AKT-mTOR pathway

Violin plots (Fig 3A–3D) showed significant upregulation of ASNS, AKT2, and MTOR, and a moderate increase in AKT3, at 48 hours post-CMV infection compared to non-infected controls, with ASNS, AKT2, and MTOR displaying statistically

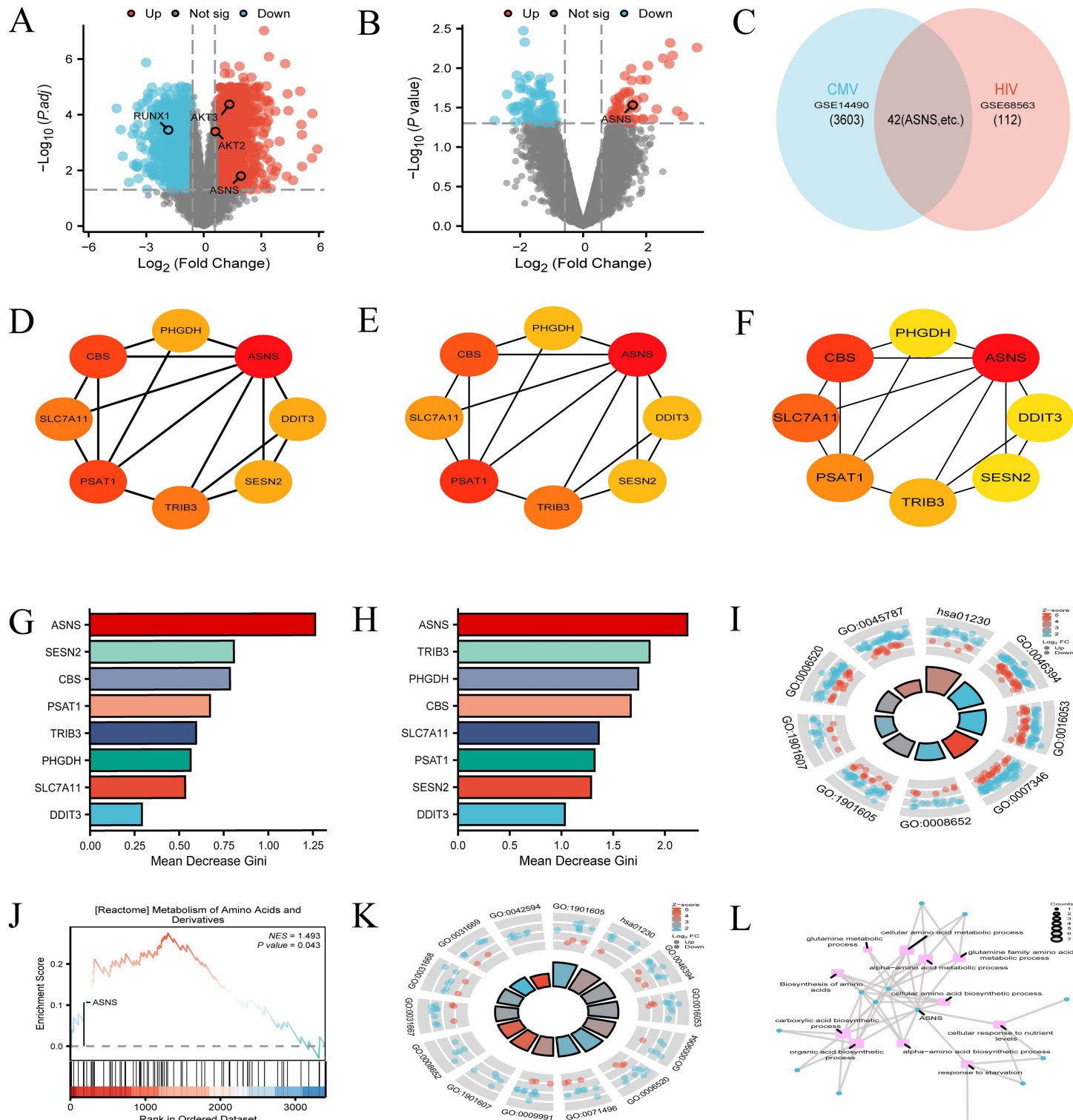

**Fig 2. ASNS expression and enrichment in CMV/HIV infections. (A)** CMV infection upregulates ASNS. **(B)** HIV-1 infection upregulates ASNS. (C) 42 genes overlap in CMV/HIV. **(D)** PPI network shows ASNS as top hub. **(E)** MCC ranks ASNS first. **(F)** Centrality analysis highlights ASNS. **(G)** Random Forest confirms ASNS influence. **(H)** ASNS has high Gini scores. **(I)** ASNS linked to biosynthesis in CMV. **(J)** GSEA shows ASNS enrichment in CMV. **(K)** ASNS associated with biosynthesis in HIV. **(L)** ASNS role in co-infections underscored.

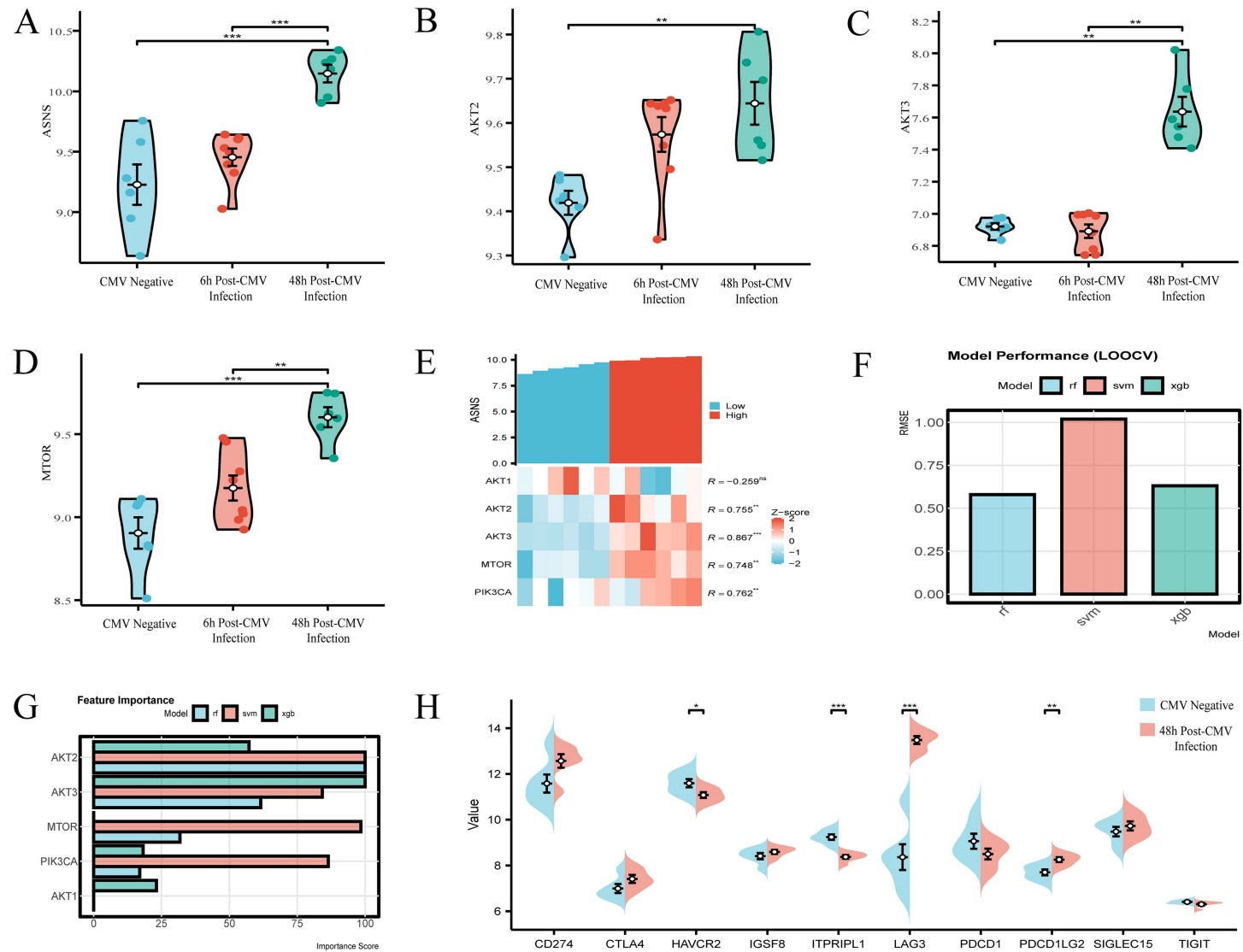

**Fig 3. ASNS and PI3K-AKT-mTOR pathway expression analysis. (A)** ASNS upregulation at 48h post-CMV infection. **(B)** AKT2 upregulation at 48h post-CMV infection. **(C)** AKT3 moderate increase at 48h post-CMV infection. **(D)** MTOR upregulation at 48h post-CMV infection. **(E)** Co-expression of ASNS with PI3K-AKT-mTOR pathway molecules. **(F-G)** Random Forest model shows AKT2 as key node. **(H)** Immune checkpoint gene expression changes post-CMV infection.

significant elevation (p < 0.001). In the GSE14490 dataset, co-expression heatmap analysis of non-infected and 48-hour post-infection samples from CMV-negative healthy blood donors revealed strong positive correlations between ASNS and key PI3K-AKT-mTOR pathway molecules in CMV-infected dendritic cells (**Fig 3E**), as determined by Spearman correlation analysis: PIK3CA (R = 0.762, p < 0.01), MTOR (R = 0.748, p < 0.01), AKT3 (R = 0.867, p < 0.001), and AKT2 (R = 0.755, p < 0.01). Random forest (RF) analysis of the same dataset (**Fig 3F**–**3G**) demonstrated optimal predictive performance with the lowest RMSE (0.58) and highest R-squared (0.64), while feature importance analysis identified AKT2 as the most significant PI3K-AKT-mTOR pathway molecule associated with ASNS (importance score 100.0 in both RF and XGB models). Expression analysis of immune checkpoint genes showed significant differential expression between non-infected

and 48-hour post-CMV infection samples (Fig 3H): ITPRIPL1 was significantly downregulated (p < 0.001), whereas LAG3 was significantly upregulated (p < 0.001). Collectively, these findings demonstrate consistent upregulation of ASNS and PI3K-AKT-mTOR pathway components, strong correlative patterns between ASNS and pathway molecules, and differential expression of immune checkpoint genes in response to CMV infection, highlighting interconnected molecular signatures relevant to viral-induced cellular responses.

## Differential expression of ASNS and PI3K-AKT-mTOR pathway components in CMV and HIV infections

Analysis of the CMV-related GSE81246 dataset showed significant upregulation of PIK3CA in CMV-positive individuals compared to those with primary CMV infection (Fig 4A), highlighting a correlative link between the PI3K pathway and chronic CMV infection, while AKT1 was significantly increased in primary CMV infection relative to CMV-positive individuals (Fig 4B), reflecting pathway activation upon initial viral exposure. In HIV-related datasets, GSE6740 analysis revealed ASNS, AKT1, AKT2, AKT3, and MTOR were significantly upregulated in acute and AIDS patients compared to HIV carriers (Fig 4C–4G): AKT1 was elevated in acute patients (Fig 4D), AKT2 and AKT3 in both acute and AIDS patients (Fig 4E–4F), and MTOR in acute (p < 0.05) and AIDS (p < 0.001) patients (Fig 4G), with LAG3 also significantly upregulated in these patient groups (Fig 4H); GSE33580 analysis showed ASNS, AKT2, AKT3, and MTOR were significantly upregulated in the HIV virologic failure group versus HIV-negative controls (p < 0.001; Fig 4I–4L). Co-expression heatmap analysis of GSE6740 (encompassing early, chronic progressive, non-progressive HIV infection, and uninfected controls) revealed consistent positive correlations between ASNS and PI3K-AKT-mTOR pathway molecules (Fig 4M): AKT3 (R = 0.695, p < 0.001), PIK3CA (R = 0.605, p < 0.01), AKT2 (R = 0.594, p < 0.01), MTOR (R = 0.534, p < 0.05), and AKT1 (R = 0.526, p < 0.05). Machine learning analysis of GSE6740 (RF, SVM, XGB models) demonstrated the RF model had optimal predictive performance (RMSE = 0.226, R-squared = 0.953; Fig 4N), with feature importance analysis identifying AKT2 as the top PI3K-AKT-mTOR pathway molecule associated with ASNS (importance score = 100.0 in RF; Fig 4O), alongside significant scores for AKT3, PIK3CA, and MTOR. For immune checkpoint genes, GSE81246 analysis showed LAG3 was significantly upregulated in individuals with symptomatic primary HCMV infection compared to HCMV-seronegative and seropositive healthy donors (p < 0.001; Fig 4P). Collectively, these findings across CMV and HIV datasets demonstrate consistent upregulation of ASNS and PI3K-AKT-mTOR pathway components, strong correlative patterns between ASNS and pathway molecules (notably AKT2), and differential LAG3 expression across viral infection stages, highlighting conserved molecular signatures relevant to CMV and HIV pathogenesis.

## Single-cell transcriptomics analysis of HIV infection

scRNA-Seq data from GSE157829 were preprocessed using the Seurat package in R, with low-quality cells filtered out to retain 3,000 high-quality cells for downstream analysis. The PCA scree plot (Fig 5A) shows the variance contribution of the top 10 principal components (with associated p-values), while the scatter plot of standardized variance versus average expression for 2,000 highly variable genes (HVGs) highlights variable genes including IGKC, S100A9, S100A8, LYZ, PF4, and IGJ (Fig 5B). UMAP visualization based on the first 15 principal components clearly discriminated between cellular populations (Fig 5C), and clustering using the top 10 marker genes facilitated annotation of distinct cell types—with IGJ, IGLL5, and MZB1 identified as high-variability genes for plasma cell annotation and clustering [17]. Comparison of cellular proportions between HIV-negative and HIV-infected groups (Fig 5D–5E) revealed significant immune landscape alterations: increased proportions of plasma cells, Paneth cells, pro-inflammatory macrophages, and activated CD8+ T cells, alongside decreased exhausted CD8+ T cells, dendritic cells, memory CD4+ T cells, and helper T cells in the HIV-infected group. Functional enrichment analysis of HVGs (Fig 5F) demonstrated elevated activity of the PI3K-AKT-mTOR signaling pathway and tumor proliferation signature in HIV-positive versus HIV-negative individuals. At the protein level, plasma cells from HIV-positive patients showed

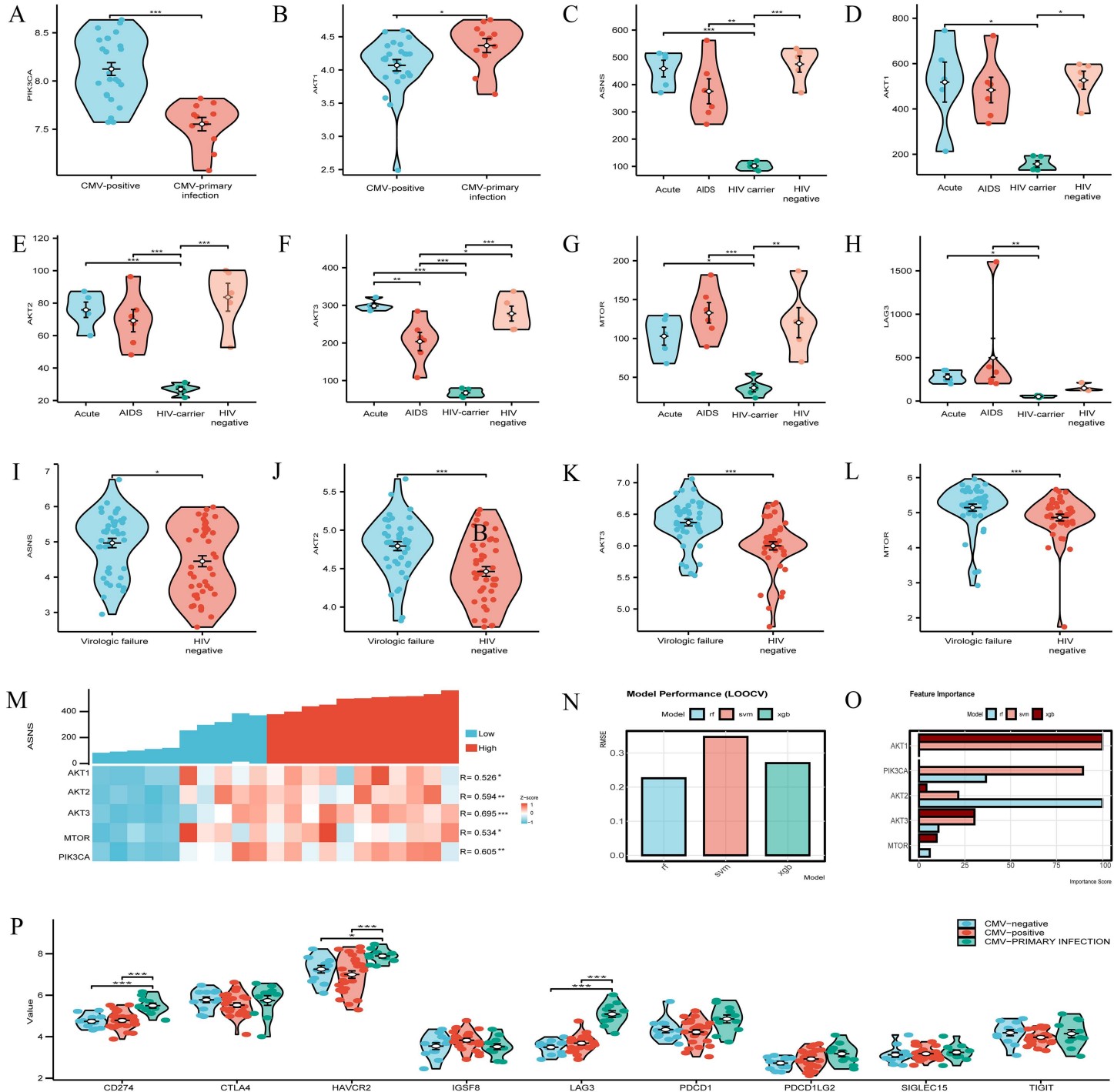

**Fig 4. ASNS and PI3K-AKT-mTOR expression in CMV/HIV infections. (A)** PIK3CA up in CMV-positive. **(B)** AKT1 up in CMV-primary. **(C)** ASNS up in acute and AIDS groups. **(D)** AKT1 up in acute HIV. **(E-H)** AKT2/3, MTOR, and LAG3 up in acute and AIDS. **(I-L)** ASNS, AKT2/3, and MTOR up in HIV virologic failure. **(M)** ASNS co-expresses with PI3K-AKT-mTOR in HIV. **(N)** Random Forest model performance in HIV. **(O)** Machine learning identifies AKT2 as key node in HIV. **(P)** LAG3 up in primary HCMV infection.

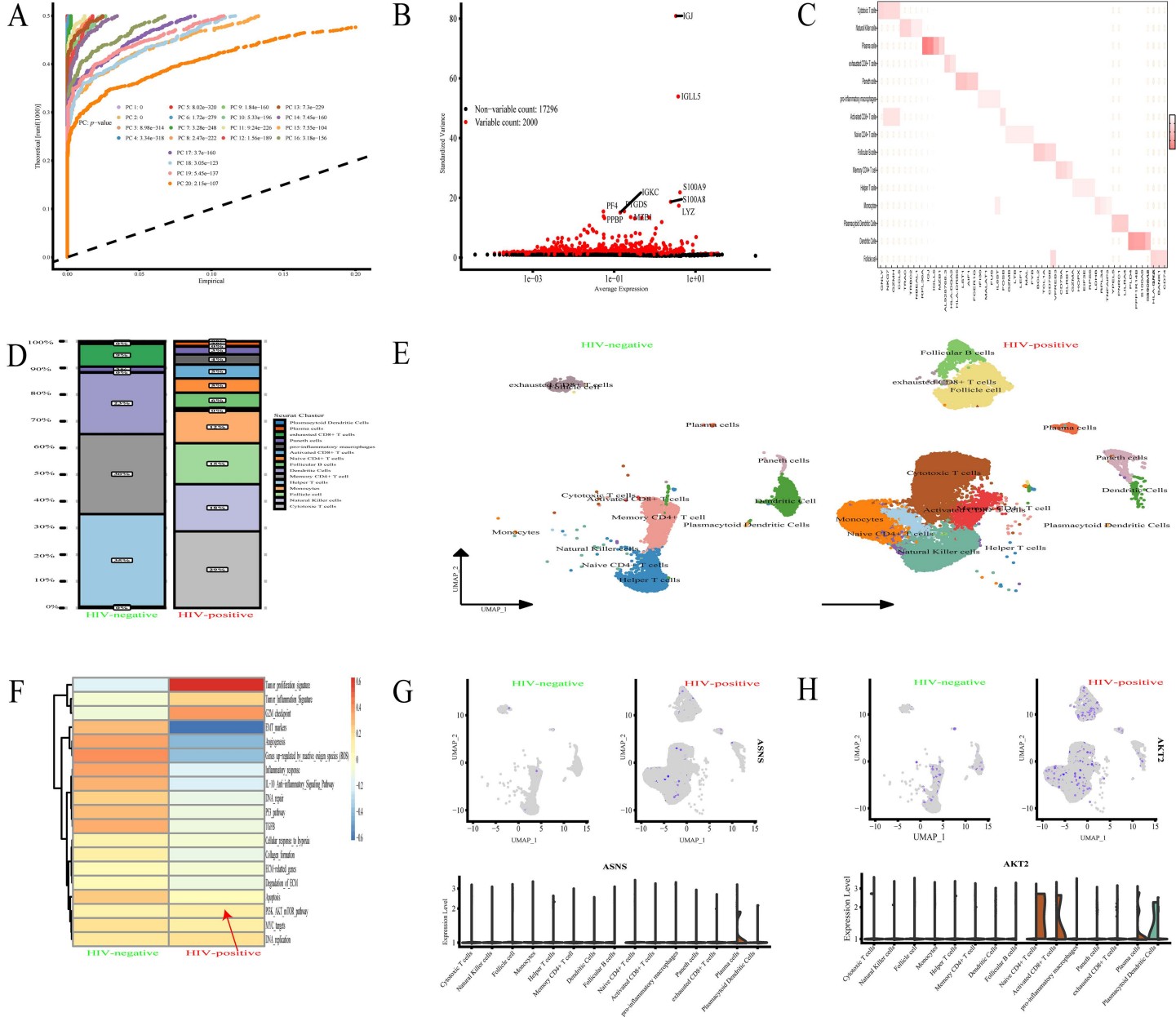

**Fig 5. Single-cell transcriptomics analysis of HIV infection. (A-C)** PCA and UMAP show cellular populations. **(D-E)** HIV alters immune cell proportions. **(F-H)** HIV upregulates PI3K-AKT-mTOR pathway and ASNS/AKT2 in plasma cells.

increased expression of ASNS (**Fig 5G**) and AKT2—a key kinase in the PI3K-AKT-mTOR pathway (**Fig 5H**)—with these findings aligning with observed correlative patterns between ASNS and the PI3K-AKT-mTOR pathway. Collectively, these R-generated scRNA-Seq analyses reveal HIV infection-induced changes in immune cell composition, HVG-associated pathway enrichment, and elevated ASNS/AKT2 expression in plasma cells, highlighting interconnected molecular and cellular signatures relevant to HIV infection.

## Transcription factors regulating ASNS expression across multiple datasets

Bioinformatics analysis predicted eight transcription factor candidates for IGJ, IGLL5, MZB1, and ASNS: ZFX, BRD4, MYB, NFKB1, SMARCA4, STAT3, RCOR1, and RUNX1 (**Fig 6A**). Random forest (RF) analysis with the Mean Decrease Gini metric in the GSE14490 dataset identified key transcription factors (**Fig 6B**), with RUNX1 showing the highest score, and ROC analysis confirmed its high predictive accuracy in CMV infection (**Fig 6C**); RF analysis of the GSE68563 dataset similarly highlighted RUNX1's relevance to HIV-1 infection (**Fig 6D**). The JASPAR database predicted RUNX1's consensus binding motif (**Fig 6E**), and the UCSC Genome Browser identified potential RUNX1 binding sites on the ASNS promoter (**Fig 6F**), indicating a potential correlative link between RUNX1 and ASNS transcriptional regulation. Analysis of the GSE81246 dataset showed higher MDM2 expression in individuals with symptomatic primary HCMV infection compared to controls (**Fig 6G**). In the GSE14490 dataset, dynamic differential expression of transcription factors was observed across CMV infection stages: ATF4 expression increased at 6 hours post-infection, and RUNX1 expression differed between early and late stages (**Fig 6H**–**6J**). In the GSE6740 dataset, ATF4, MDM2, and RUNX1 were downregulated in HIV carriers relative to other groups (**Fig 6K**–**6M**), while in the GSE33580 dataset, these transcription factors were upregulated in the HIV-resistant group compared to controls (**Fig 6N**–**6P**). UMAP visualizations from the GSE157829 dataset highlighted MDM2's specific expression in plasma cells of HIV patients (**Fig 6Q**–**6R**). Collectively, these findings identify RUNX1 as a consistently relevant transcription factor across CMV and HIV datasets, with potential binding sites on the ASNS promoter, and reveal differential expression patterns of ATF4, MDM2, and RUNX1 across viral infection stages and clinical groups, highlighting correlative signatures relevant to ASNS transcriptional regulation in CMV and HIV infection.

## GSE33580 dataset analysis identifies RUNX1 as the key biomarker for HIV resistance with moderate diagnostic and clinical utility

The GSE33580 dataset comprises 86 samples (43 HIV-resistant individuals and 43 HIV-negative controls) and 20,549 genes. QC analysis confirmed high data integrity: missing values were 0%, zero-variance genes accounted for 0%, median sample CV was 0, and median pairwise sample correlation was 0.928. Log2-transformed expression values ranged from 0.06 to 18.53. The normalized expression distribution, characteristic of high-quality microarray data, is shown in **Fig 7A**. **Fig 7B** shows normalized average feature importance across three models: RUNX1 scored a maximum 1.000—nearly 1.4-fold higher than second-ranked MDM2 (0.703)—with ATF4 (0.615), ASNS (0.408), and AKT2 (0.279) declining in importance, confirming RUNX1 as the most influential feature. ROC analysis (**Fig 7C**) identifies RUNX1 as the sole biomarker with moderate diagnostic utility (AUC = 0.714), while MDM2 (AUC = 0.58), ASNS (AUC = 0.534), AKT2 (AUC = 0.524), and ATF4 (AUC = 0.493) have minimal to no discriminative power. **Fig 7D**'s clinical impact curve demonstrates that at clinically relevant thresholds (0.2–0.6), identified high-risk individuals align proportionally with actual events, reflecting the model's ability to stratify risk without excessive false positives. **Fig 7E**'s decision curve analysis shows the model delivers positive net benefit across 0.1–0.7 thresholds, outperforming "treat all" and "treat none" strategies in this clinically meaningful range. **Fig 7F**'s clinical prediction nomogram integrates all five features (RUNX1 contributing the highest point weight), and its performance—quantified by an AUC of 0.737—reflects moderate yet robust predictive ability to distinguish HIV-resistant from HIV-negative individuals.

## LDA-Based RUNX1 model: Optimal diagnostic for HIV treatment resistance

The top-performing diagnostic model pairs LDA (Linear Discriminant Analysis) with the single gene RUNX1, achieving an AUC of 0.728—the highest among all algorithm-gene combinations (**Fig 8C**). This model delivers balanced performance across key clinical metrics: accuracy (0.622), sensitivity (0.535), specificity (0.709), and F1 score (0.586) (**Fig 8A**), with no critical gaps in real-world applicability. Decision Tree, by contrast, excels in sensitivity (mean = 0.791, **Fig 8A**) but lacks specificity (0.593) and peaks at a lower AUC (0.715 with the RUNX1 + MDM2 pair, **Fig 8C**), leading to unbalanced diagnostic performance.

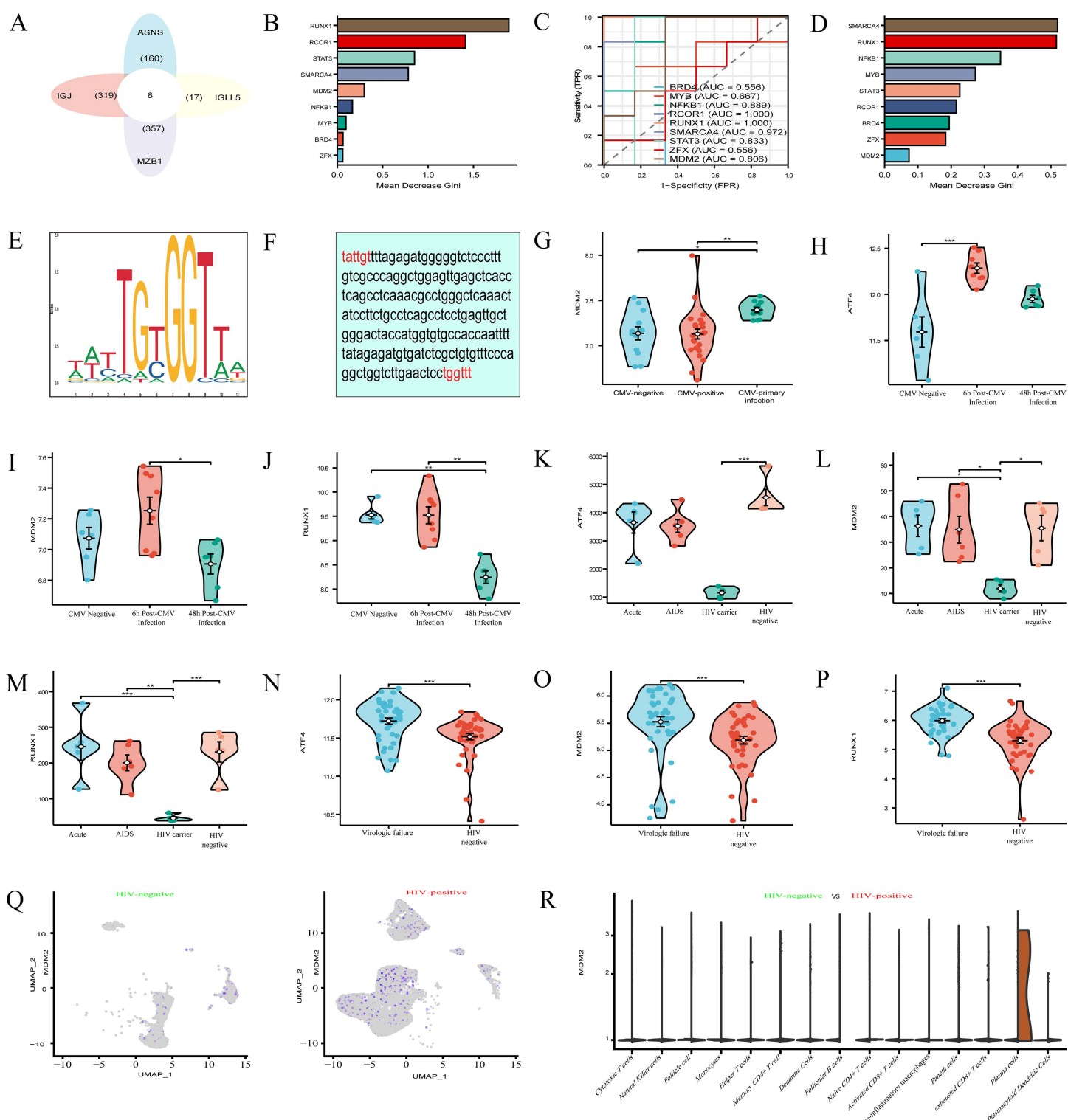

**Fig 6. Transcription factors regulating ASNS in CMV and HIV infections. (A)** Predicted transcription factors for IGJ, IGLL5, MZB1, and ASNS. **(B)** RUNX1 key in CMV. **(C)** RUNX1 accuracy in CMV. **(D)** RUNX1 importance in HIV-1. **(E)** RUNX1 motif prediction. **(F)** RUNX1 binding sites on ASNS promoter. **(G)** MDM2 up in HCMV. **(H-J)** ATF4, MDM2, RUNX1 dynamics across CMV stages. **(K-M)** ATF4, MDM2, RUNX1 down in HIV carriers. **(N-P)** ATF4, MDM2, RUNX1 up in HIV-resistant. **(Q-R)** MDM2 in HIV plasma cells.

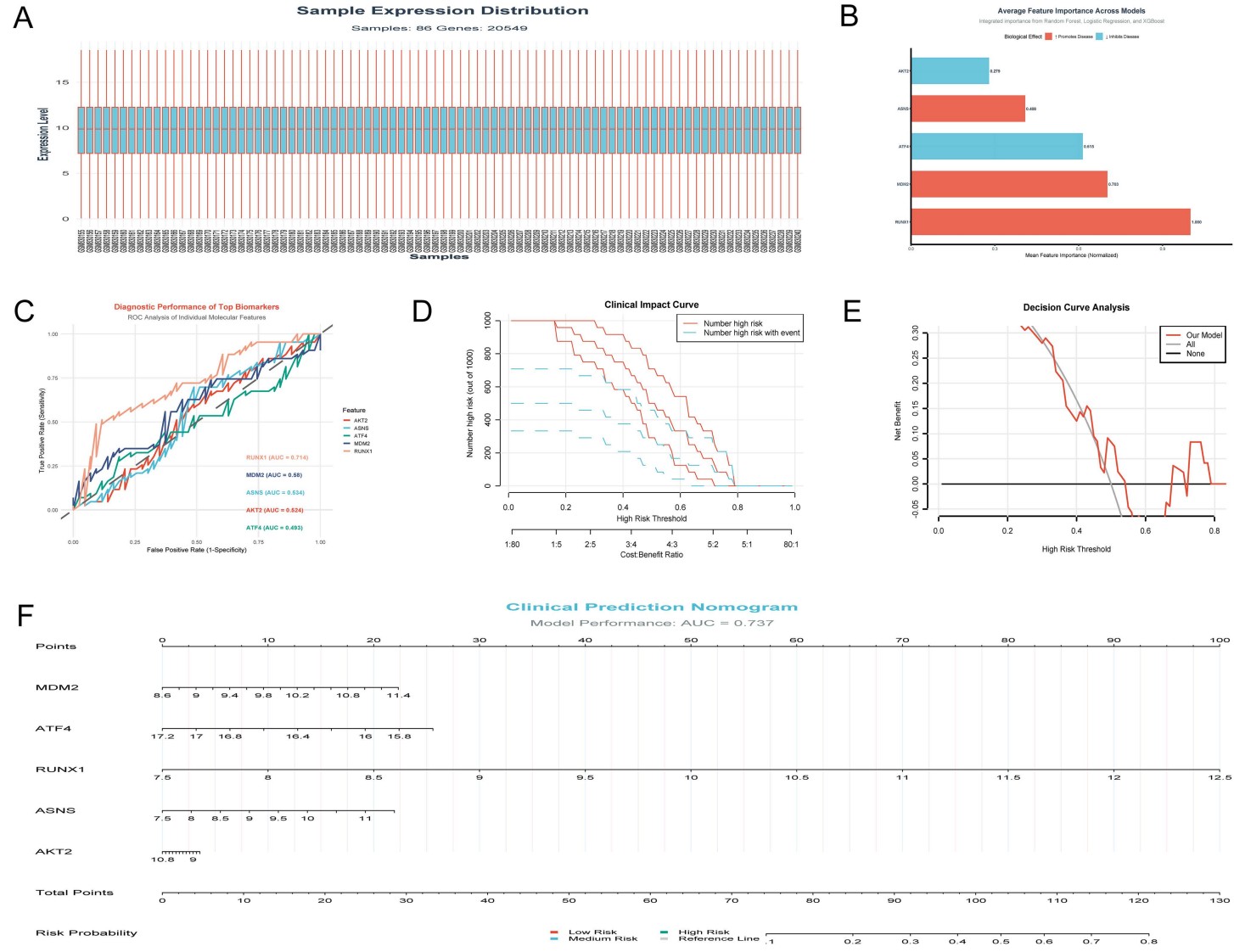

**Fig 7. Validation of RUNX1 as a key biomarker for HIV resistance in GSE33580. (A)** Normalized expression distribution of GSE33580 (high-quality microarray data); **(B)** Normalized avg feature importance (3 models), confirming RUNX1's top influence; **(C)** ROC analysis of candidates: RUNX1 as sole moderate diagnostic marker (AUC = 0.714); **(D)** Clinical impact curve: risk stratification without excessive false positives at 0.2–0.6 thresholds; **(E)** Decision curve analysis: positive net benefit at 0.1–0.7 thresholds (outperforming "treat all"/"treat none"); **(F)** Clinical prediction nomogram (5 features, RUNX1 highest weight): nomogram's moderate diagnostic performance (AUC = 0.737) for HIV-resistant/negative distinction.

Fig 8B (PCA biplot) provides insights into potential relationships between molecules: the relative angles between feature vectors indicate their pairwise correlations. RUNX1, MDM2, and ATF4 exhibit small to moderate angles between their vectors, suggesting weak to moderate positive correlations. In contrast, ASNS and AKT2 form larger angles with RUNX1, MDM2, and ATF4, pointing to weaker or no significant correlation with these three molecules. Additionally, RUNX1's vector length and position confirm it as a key driver of sample clustering along the primary components (Dim1: 34.6%, Dim2: 23%), aligning with its consistent performance across top models. Expanding gene combinations (2–5 genes) failed to outperform the single-RUNX1 model, with AUC values ranging from 0.627 to 0.718 (Fig 8C). LDA's superior AUC and balanced multi-metric profile establish it as the preferred algorithm, while RUNX1 emerges as the standalone optimal diagnostic molecule.

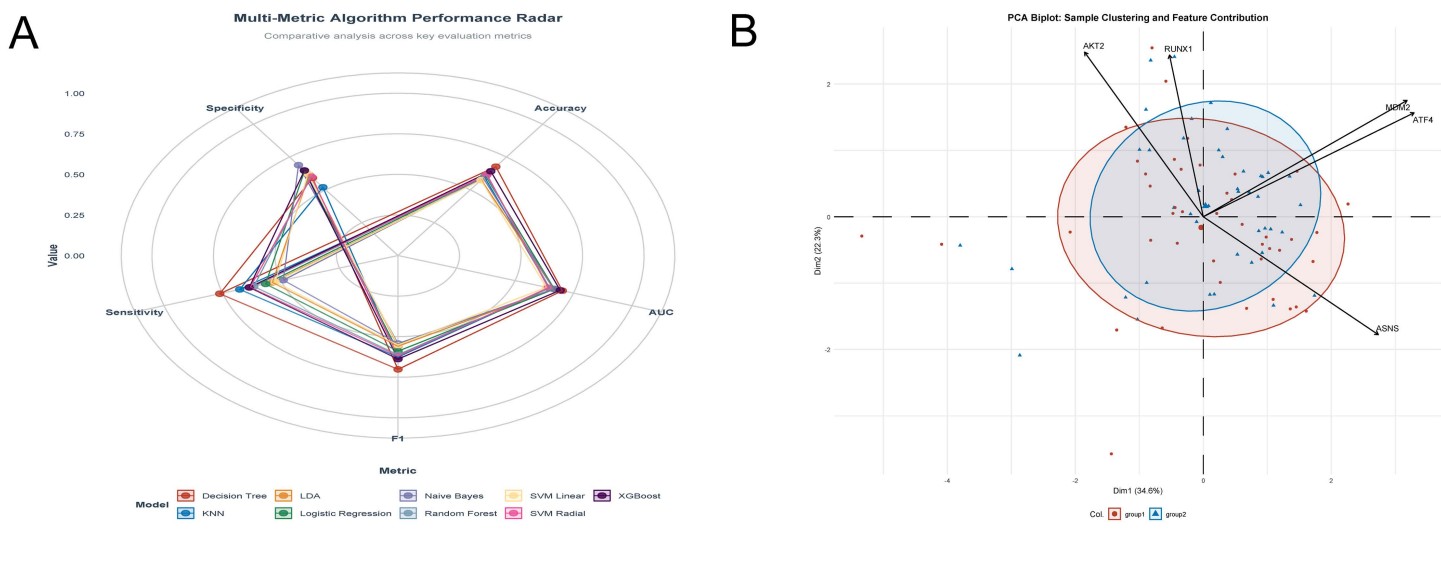

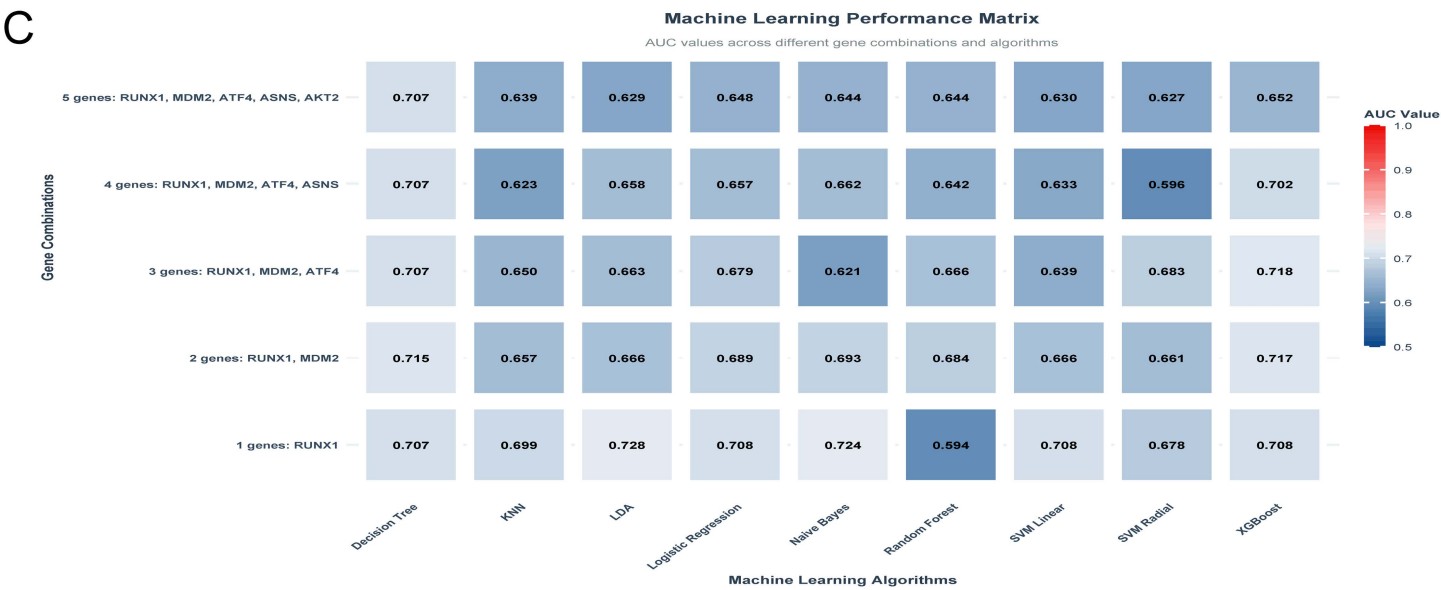

**Fig 8. LDA Model with RUNX1 for HIV Treatment Resistance Diagnosis and RUNX1-Mediated Clustering (A) Optimal LDA model (RUNX1):** balanced clinical metrics (accuracy = 0.622, sensitivity = 0.535, specificity = 0.709, F1 = 0.586); (B) PCA biplot: RUNX1 as key driver of sample clustering (Dim1:34.6%, Dim2:23%); (C) AUC comparison: LDA model with highest AUC = 0.728.

## Binding affinity and interaction analysis of ASNS protein with small molecules

Using the LeDock program, redocking of ONL (**Fig 9A**) formed 5 hydrogen bonds with ASNS at residues including GLN-59 and TYR-73, with a binding energy of −4.25 kcal/mol and RMSD of 0.02—confirming the protocol's ability to reproduce the original binding conformation. Bisabosqual A (**Fig 9B**) formed 3 hydrogen bonds at sites such as GLN-59 and ARG-48 (binding energy: −2.07 kcal/mol; RMSD: 1.24), while cidofovir (**Fig 9C**) exhibited the strongest binding affinity among the three compounds, forming 9 hydrogen bonds at residues including LEU-49 and ARG-142 (binding energy: −6.61 kcal/mol; RMSD: 0.119); all three ligands shared a common binding site at VAL-51 on ASNS (**Fig 9D**). With AutoDock Vina, ONL (**Fig 9E**) formed hydrogen bonds with VAL-52, VAL-51, GLU-76,

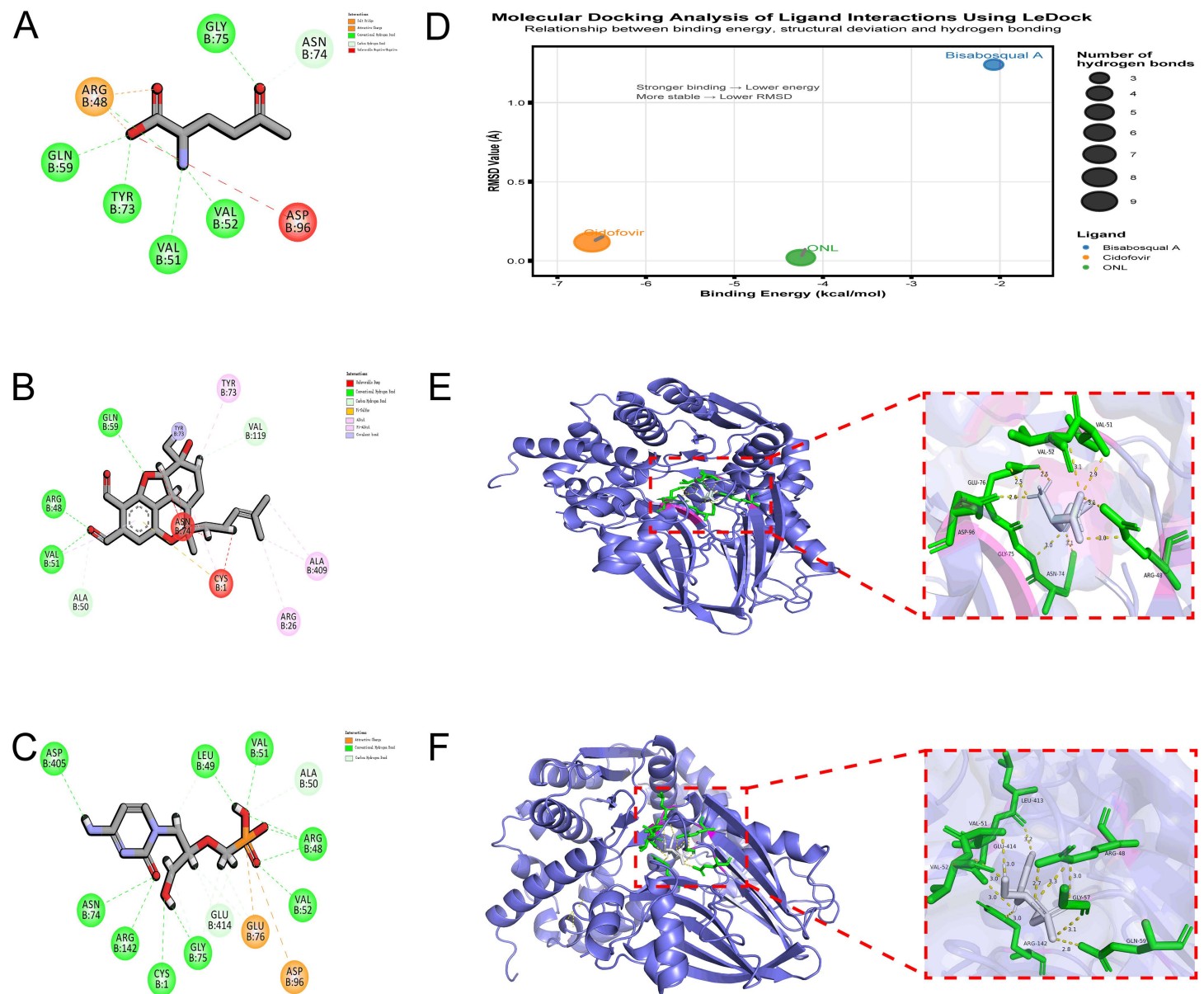

**Fig 9. ASNS small molecule interactions. (A)** ONL forms 5 hydrogen bonds with ASNS. **(B)** Bisabosqual A forms 3 hydrogen bonds with ASNS. **(C)** Cidofovir forms 9 hydrogen bonds with ASNS. **(D)** Binding energy and RMSD for ligands. **(E)** ONL's binding mode with ASNS residues. **(F)** Cidofovir's binding mode with ASNS residues.

ASP-96, GLY-75, ASN-74, and ARG-48 (binding energy: −5.7 kcal/mol; RMSD: 0.519 relative to the original PDB structure), and cidofovir (**Fig 9F**) interacted with LEU-413, VAL-51, GLU-414, VAL-52, ARG-48, GLY-57, ARG-142, and GLN-59 (binding energy: −4.2 kcal/mol). Collectively, these molecular docking results demonstrate that ONL, bisabosqual A, and cidofovir bind to ASNS via distinct hydrogen bond patterns, with cidofovir showing the highest binding affinity in LeDock analysis, and VAL-51, VAL-52, and ARG-142 emerging as shared interaction residues across ligands.

## Molecular dynamics analysis of ASNS-Ligand complexes

Molecular dynamics (MD) simulations over 200 ns revealed distinct stability profiles for the ASNS-ONL and ASNS-cidofovir complexes (**Fig 10A**–**10J**). RMSD analysis showed the ASNS-cidofovir complex exhibited greater stability with fewer fluctuations compared to the ASNS-ONL complex (**Fig 10A**), while RMSF analysis indicated higher flexibility in the ASNS-ONL complex and lower fluctuations (more stable protein structure) in the ASNS-cidofovir complex (**Fig 10B**). Center of mass (COM) distance analysis demonstrated the ASNS-cidofovir complex maintained a more consistent COM distance, reflecting robust binding interactions (**Fig 10C**), and radius of gyration (RG) analysis confirmed both complexes were stable, with the ASNS-cidofovir complex showing a lower RG value indicative of a more compact structure (**Fig 10D**). Solvent-accessible surface area (SASA) analysis revealed stable profiles for both complexes, with the ASNS-cidofovir complex exhibiting higher buried SASA values suggestive of stronger binding (**Fig 10E**). Hydrogen bond analysis showed the ASNS-cidofovir complex formed a greater number of hydrogen bonds than the ASNS-ONL complex (**Fig 10F**). Two-dimensional (2D) representations of the final simulation conformations highlighted distinct molecular interaction patterns, with the ASNS-cidofovir complex displaying a more extensive network of stabilizing hydrogen bonds (**Fig 10G**, **10I**). Hydrogen bond frequency analysis identified residue ASN-74 as critical for forming stable hydrogen bonds with both ligands, with a more prominent role in the ASNS-cidofovir complex (**Fig 10H**, **10J**). Collectively, these MD simulation results demonstrate that the ASNS-cidofovir complex exhibits superior stability compared to the ASNS-ONL complex, characterized by reduced structural fluctuations, more consistent binding interactions, a more compact conformation, higher buried SASA, and an increased number of hydrogen bonds—with ASN-74 emerging as a key residue mediating ligand-ASNS interactions.

## Discussion

Notably, this is the first study to identify ASNS as a core metabolic-signaling hub in the pathogenesis of both CMV and HIV infections through integrated multi-dataset analyses. Our findings consistently demonstrate significant ASNS upregulation in response to both viruses, consistent with previous reports that ASNS modulates viral replication and immune evasion by regulating cellular protein synthesis. Crucially, ASNS was consistently enriched in the "amino acid biosynthesis pathway (hsa01230)" across both CMV and HIV datasets, reinforcing its function as a key metabolic regulator linking viral infection to host metabolic reprogramming [7,8]. Beyond transcriptional elevation, PPI network analysis further established ASNS as a top hub gene among shared DEGs between CMV and HIV datasets. This central position in conserved molecular networks implies ASNS may mediate common pathogenic processes underlying both infections—filling a critical gap in our understanding of coinfection-specific mechanisms, as most prior research has focused on CMV or HIV in isolation. Additionally, our identification of RUNX1 as a key transcription factor regulating ASNS (with predicted binding sites on the ASNS promoter) and as the top biomarker for HIV treatment resistance (AUC = 0.714) underscores a potential "ASNS-RUNX1 regulatory axis" with both mechanistic depth and clinical implications. Collectively, these observations support the hypothesis that ASNS represents a conserved host vulnerability exploited by both viruses to facilitate infection.

Our analyses underscore a strong correlative relationship between ASNS and the PI3K-AKT-mTOR pathway—a central signaling cascade hijacked by numerous viruses to support replication and survival [18]. Through co-expression analysis and machine learning, AKT2 emerged as a key pathway molecule associated with ASNS across multiple independent datasets (e.g., GSE14490, GSE6740), with robust positive correlations also observed between ASNS and other pathway components (e.g., PIK3CA, MTOR). This consistency across datasets strengthens the potential biological significance of the ASNS-AKT2 interaction. These patterns may point to a potential reciprocal regulatory loop: ASNS upregulation could activate the PI3K-AKT-mTOR pathway, with AKT2 potentially enhancing ASNS expression in return—creating a self-sustaining axis that rewires cellular metabolism to favor viral propagation. Such metabolic reprogramming, driven by ASNS-pathway crosstalk, could boost nutrient uptake and metabolic flux, generating a cellular environment conducive to the replication of both CMV and HIV. Importantly, this proposed regulatory axis aligns with the known role of the

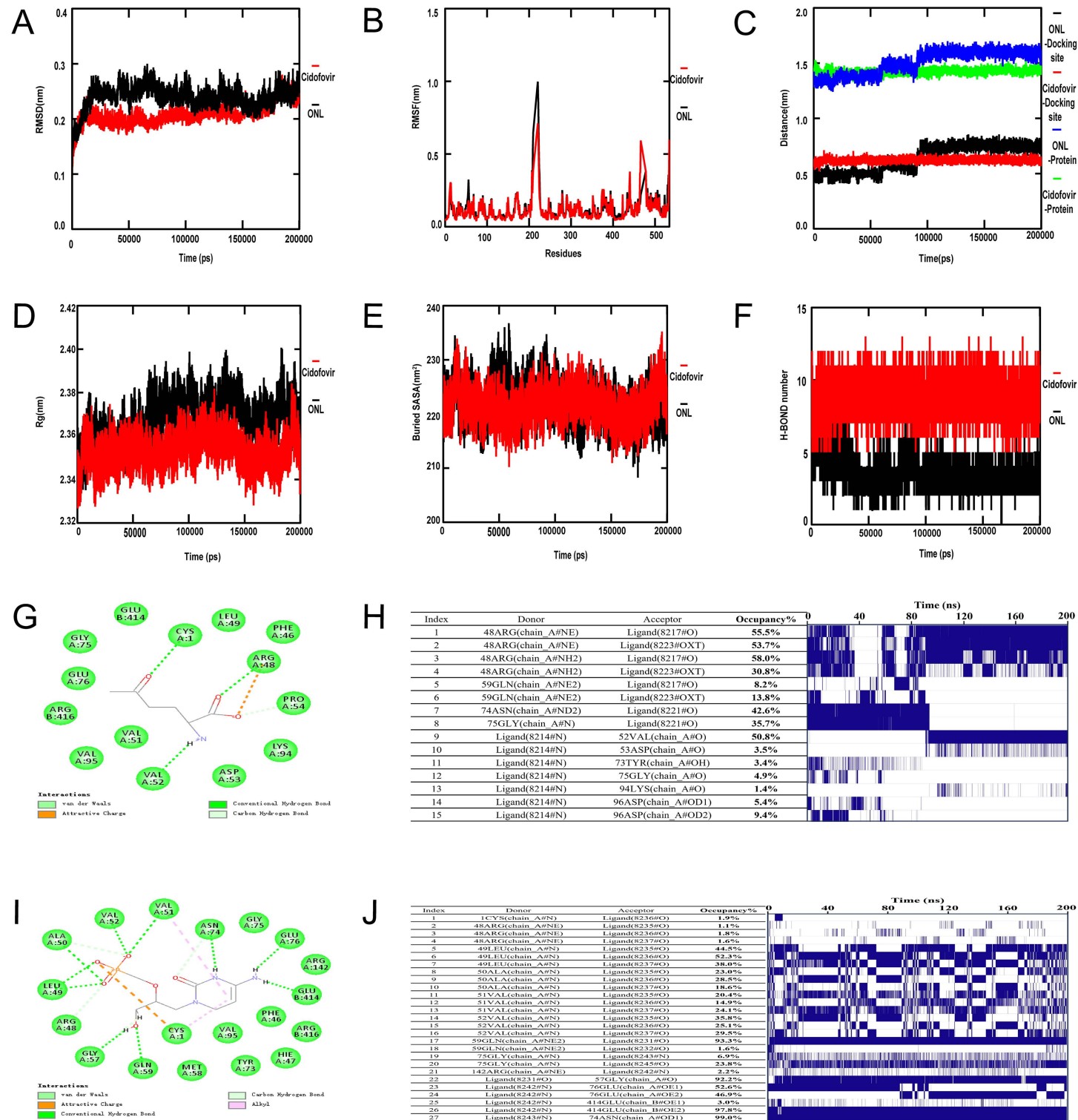

**Fig 10. Molecular dynamics of ASNS-ligand complexes. (A)** RMSD stability. **(B)** RMSF fluctuation. **(C)** COM distance consistency. **(D)** RG compactness. **(E)** SASA buried surface area. **(F)** Hydrogen bond formation. **(G and I)** Interaction diagrams of ASNS with ONL (G) and cidofovir **(I)**. **(H and J)** Role of ASN-74 in hydrogen bonding for ASNS-ONL (H) and ASNS-cidofovir **(J)**.

PI3K-AKT-mTOR pathway in viral pathogenesis [19,20], suggesting ASNS may act as an upstream modulator of this pathway to facilitate viral hijacking of host metabolic resources [8].

Significant differential expression of immune checkpoint genes (e.g., LAG3 upregulation, ITPRIPL1 downregulation) was observed following CMV infection, and these changes may be linked to the ASNS-PI3K-AKT-mTOR pathway axis. Specifically, LAG3 upregulation—an immune checkpoint associated with T cell exhaustion [21,22]—could be a downstream consequence of ASNS-mediated pathway activation, potentially contributing to viral immune escape by dampening antiviral immune responses. This cross-talk between metabolic signaling (ASNS-PI3K-AKT-mTOR) and immune checkpoint regulation highlights a coordinated viral strategy to subvert both host metabolism and immune surveillance.

Single-cell RNA sequencing (scRNA-seq) analysis further revealed specific ASNS upregulation in plasma cells from HIV-positive individuals. While the analyzed HIV patients were not explicitly coinfected with CMV, this cell-specific expression pattern may hint at a potential role for plasma cells in CMV susceptibility among HIV-infected populations—given the high prevalence of CMV coinfection in this group [23]. However, this association remains speculative without direct coinfection data, and future studies are needed to validate whether plasma cell-derived ASNS contributes to CMV reactivation or pathogenesis in coinfected individuals. Collectively, these findings emphasize the value of cell-type-specific analysis in unraveling ASNS's role in viral infections and immune regulation [7].

The molecular docking results revealing high-affinity binding between cidofovir (a clinically used antiviral) [24] and ASNS are particularly intriguing, as they may indicate potential off-target effects of existing antiviral agents on host metabolic enzymes. The identification of shared interaction residues (e.g., VAL-51, ASN-74) across ligands further provides a preliminary framework for rational drug design targeting ASNS. However, these in silico predictions require in vitro and in vivo validation to confirm functional relevance, and overinterpreting their clinical applicability is premature at this stage.

While this study provides novel insights, it is not without limitations. First, reliance on public GEO datasets constrains the results to the specific cell types and patient cohorts included, which may not fully represent the diversity of CMV/HIV-infected individuals. Second, the absence of direct coinfection patient data limits our ability to fully elucidate molecular interactions between CMV and HIV in a clinically relevant setting. Third, all conclusions are based on correlative analyses, and functional validation of key predictions (e.g., ASNS-pathway interactions, RUNX1-mediated ASNS regulation) is essential to confirm causal relationships. Fourth, while the diagnostic potential of the RUNX1-ASNS axis is promising, it requires validation in larger, more diverse cohorts to determine its clinical utility.

To address these limitations and build on our findings, several future research avenues are warranted. First, multi-omics integration (transcriptomics, proteomics, metabolomics) would provide a more holistic understanding of how ASNS mediates CMV/HIV pathogenesis, validating protein expression levels and metabolic changes associated with ASNS upregulation. Second, functional experiments (e.g., ASNS knockdown/overexpression in viral infection models) are needed to confirm ASNS's role and its interaction with the PI3K-AKT-mTOR pathway in viral replication. Third, direct analysis of coinfection patient samples would enable validation of the proposed ASNS-mediated mechanisms in a clinically relevant context. Fourth, preclinical studies could explore the therapeutic potential of targeting ASNS or its interaction with AKT2—particularly in combination with existing antivirals or immune checkpoint inhibitors. Fifth, exploring the clinical utility of the RUNX1-ASNS axis as a diagnostic marker for HIV treatment resistance could inform personalized therapeutic approaches. Finally, further investigation into ASNS's transcriptional regulation (e.g., RUNX1 binding to the ASNS promoter) could uncover additional regulatory nodes with therapeutic potential.

In conclusion, our integrated bioinformatics analyses identify ASNS as a conserved metabolic-signaling hub in CMV and HIV infections—tied to amino acid metabolism, the PI3K-AKT-mTOR pathway, and RUNX1-driven regulation—with implications for developing novel therapeutic and diagnostic strategies targeting viral hijacking of host metabolism. The proposed crosstalk between ASNS and the PI3K-AKT-mTOR pathway, along with its links to immune evasion and cell-specific expression, provides a framework for future functional and clinical studies to advance our understanding and management of CMV/HIV coinfection.

## Conclusions

In conclusion, we are the first to identify ASNS as a metabolic-signaling hub in CMV/HIV coinfection via integrated bioinformatics, aligning with viral metabolic hijacking—wherein pathogens exploit host networks for replication and immune evasion—while illuminating shared mechanisms. Our findings underscore ASNS's central role in coinfection: consistently upregulated in both viruses, it acts as a top PPI hub, shares strong correlations with the PI3K-AKT-mTOR pathway (notably cross-validated AKT2), and is enriched in amino acid biosynthesis—linking infection to metabolic reprogramming. Transcriptional analyses identify RUNX1 as a key ASNS modulator (predicted promoter binding) and top HIV treatment resistance biomarker, underscoring the ASNS-RUNX1 axis's clinical relevance, while scRNA-seq reveals ASNS upregulation in HIV-positive plasma cells—implicating cell-type-specific metabolic dysregulation. Molecular docking/simulations show preliminary high-affinity cidofovir-ASNS binding, pointing to potential off-target metabolic modulation worthy of further exploration. Limited to correlative and in silico analyses, these findings provide a new coinfection framework, nominate ASNS as a promising experimental target, and hint at potential for host-directed therapy—with future research prioritizing ASNS pathway validation, multi-omics for mechanistic insights, and clinical evaluation of the ASNS-RUNX1 axis in larger cohorts.

## Supporting information

**S1 Table. Ligands and binding data for ASNS interactions.**
(XLS)

**S2 Table. DEGs in CMV and HIV infections.**
(XLSX)

**S3 Table. Predicted transcription factors for ASNS.**
(XLSX)

## Author contributions

**Conceptualization:** JunKai Sun, HaoGang Zhu, Wei Lu, Ye Fang.

**Data curation:** ShaoXiang Ding, HongXia Bao, WenJun Chen, Bo Cai.

**Formal analysis:** Hao Zhang, ShuYou Yuan, HongXia Bao, WenJun Chen, Bo Cai.

**Funding acquisition:** JunKai Sun, Wei Lu, Ye Fang.

**Investigation:** Wei Lu.

**Methodology:** ShaoXiang Ding, WenJun Chen, Bo Cai, Wei Lu.

**Project administration:** HaoGang Zhu, Ye Fang.

**Resources:** Wei Lu, Ye Fang.

**Supervision:** JunKai Sun, HaoGang Zhu, Wei Lu, Ye Fang.

**Validation:** HaoGang Zhu.

**Writing – original draft:** Hao Zhang, ShuYou Yuan.

**Writing – review & editing:** JunKai Sun, HaoGang Zhu, Wei Lu, Ye Fang.

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
