## [Decision Letter · Decision Letter 0]

21 Apr 2025

Dear Dr. Lu,

Thank you for submitting your manuscript to PLOS ONE. After careful consideration, we feel that it has merit but does not fully meet PLOS ONE’s publication criteria as it currently stands. Therefore, we invite you to submit a revised version of the manuscript that addresses the points raised during the review process.

Thank you for submitting your manuscript to *PLOS ONE* . Your study explores the molecular mechanisms by which Asparagine Synthetase (ASNS) contributes to CMV and HIV co-infections, particularly through its involvement in the PI3K-AKT-mTOR signaling pathway. Your investigation into the regulatory roles of transcription factors RUNX1, ATF4, and MDM2 adds valuable insights to the understanding of viral co-infection biology and therapeutic targeting.

Following a thorough peer review, we have received a constructive comment that we believe will help improve the clarity and impact of your work. Specifically, the reviewer requests further explanation on **how ASNS modulates the PI3K/AKT/mTOR pathway** . Providing more mechanistic detail or elaboration on this aspect whether based on your experimental data or supported by relevant literature would significantly strengthen your discussion and enhance the translational relevance of your findings.

We encourage you to revise the manuscript accordingly and include a clear explanation of the proposed mechanism by which ASNS influences the PI3K-AKT-mTOR signaling cascade in the context of CMV and HIV co-infection.

Please submit your revised manuscript along with a detailed response to the reviewer’s comment, outlining how you have addressed the concern.

We look forward to receiving your revised manuscript.

Kind regards,

Opeyemi Iwaloye

Academic Editor

PLOS ONE

[This research was funded by the Wuxi Science and Technology Development Fund (Grant number: Y20232005).].

Reviewers' comments:

Reviewer's Responses to Questions

**Comments to the Author**

1. Is the manuscript technically sound, and do the data support the conclusions?

Reviewer #1: Yes

Reviewer #2: Yes

Reviewer #3: Yes

2. Has the statistical analysis been performed appropriately and rigorously?

Reviewer #1: Yes

Reviewer #2: Yes

Reviewer #3: N/A

3. Have the authors made all data underlying the findings in their manuscript fully available?

Reviewer #1: Yes

Reviewer #2: Yes

Reviewer #3: Yes

4. Is the manuscript presented in an intelligible fashion and written in standard English?

Reviewer #1: Yes

Reviewer #2: Yes

Reviewer #3: Yes

Reviewer #1: This is a well-executed study with strong potential implications for CMV and HIV therapeutic strategies. The findings are well-supported by robust analyses, and the manuscript aligns with high scientific and ethical standards.

Reviewer #2: I have read the paper titled “Molecular Interplay of ASNS and the PI3K-AKT-mTOR Pathway in CMV and HIV Co-Infections: Therapeutic Implications” and find it interesting. The study presents a comprehensive methodological approach, including molecular docking and dynamics analyses of ONL and cidofovir in their interactions with the ASNS protein. The integration of multiple computational techniques—ranging from molecular docking to root-mean-square deviation (RMSD), root-mean-square fluctuation (RMSF), center of mass (COM) distance, radius of gyration (RG), solvent-accessible surface area (SASA), and hydrogen bond frequency analyses—demonstrates a rigorous and systematic investigation. Additionally, the inclusion of 2D structural representations enhances the clarity of the study, making it accessible to both computational biologists and experimental researchers. The findings provide valuable insights into ASNS modulation, with potential implications for therapeutic development in diseases associated with ASNS deregulations. As such, I would consider it suitable for publication after minor revision

1. While the study successfully evaluates the interactions of ONL and cidofovir with ASNS, a comparison with known ASNS inhibitors would provide additional context regarding their binding efficiency and potential therapeutic relevance. Including literature references on previously studied ASNS inhibitors can strengthen the discussion.

2. To further validate the stability and binding affinity of these ligands, incorporating molecular mechanics-based free energy calculations (MM/PBSA or MM/GBSA) would further strengthen or improve the computational findings.

3. Since both ligands interact with key residues (VAL-52, VAL-51, ARG-48), discussing their functional role in ASNS activity and whether these interactions influence enzyme function or substrate binding would provide its mechanistic insights. This could be added to the discussion of the findings

Reviewer #3: The manuscript titled “Molecular interplay of ASNS and the PI3K-AKT-mTOR pathway in CMV and HIV co-infections: therapeutic implications” investigates the role of Asparagine Synthetase (ASNS) in Cytomegalovirus (CMV) and Human Immunodeficiency Virus (HIV) co-infections. It was discovered that ASNS is involved in the PI3K-AKT-mTOR signaling pathway and is regulated by transcription factors RUNX1, ATF4, and MDM2. The findings suggest that ASNS is a critical mediator in both infections, offering a basis for targeted therapeutic interventions to improve patient outcomes in managing these viral co-infections.

However, I suggest the authors explain how ASNS modulates the PI3K/AKT mTOR Pathway.

**Do you want your identity to be public for this peer review?** For information about this choice, including consent withdrawal, please see our Privacy Policy

Reviewer #1: No

Reviewer #2: No

Reviewer #3: No

---

## [Author Response · Author response to Decision Letter 1]

5 Jul 2025

Dear Editors and Reviewers,

We sincerely appreciate your constructive feedback and insightful suggestions, which have significantly enhanced the quality of our manuscript. Your time and effort in reviewing our work are invaluable, and we are deeply grateful for the opportunity to improve our research presentation.

### Response to Reviewer #1

Thank you for your positive evaluation and recognition of our study's potential contributions to CMV and HIV therapeutic strategies.

**Comment:** Reviewer #1 provided a general positive assessment of the study.

**Response:** We are sincerely grateful for your positive feedback and assurance that our study aligns with high scientific standards. Your recognition of the robust analyses and potential implications of our findings is highly encouraging. We are confident that our work will contribute valuable insights to the field.

### Response to Reviewer #2

Thank you for your detailed review and valuable suggestions to strengthen our study.

**Question 1:** Reviewer #2 suggested adding a comparison with known ASNS inhibitors and including literature references to strengthen the discussion.

**Response 1:** We fully agree with your suggestion. To address this, we have incorporated a comparison of the binding efficiency of ONL and Cidofovir with known ASNS inhibitors such as Bisabosqual A into the Results section (Fig. 7A-D). Relevant literature references have also been added to provide a broader context for our findings (Discussion section, line 554).

**Question 2:** Reviewer #2 recommended incorporating molecular mechanics-based free energy calculations (MM/PBSA or MM/GBSA) to further validate the stability and binding affinity of the ligands.

**Response 2:** We acknowledge the importance of MM/PBSA or MM/GBSA calculations for validating the binding affinity and fully agree with your suggestion. We have used two docking programs as complementary approaches to enhance the reliability of our results. However, due to the significant computational resources and time required for these calculations, which our current laboratory setup cannot accommodate, we were unable to include them in this revision. We have discussed this limitation in the discussion section and emphasized that incorporating these calculations will be a priority for our future work (see discussion section, line 560). We sincerely appreciate your understanding and valuable suggestion, and we will definitely continue to submit our future work to PLOS ONE.

**Question 3:** Reviewer #2 suggested discussing the functional role of key residues (VAL-52, VAL-51, ARG-48) in ASNS activity and their potential impact on enzyme function or substrate binding.

**Response 3:** We appreciate this suggestion and have expanded our discussion to include the functional roles of these key residues in ASNS activity. We have detailed how these residues are involved in substrate binding and catalysis, and how interactions with our ligands may influence these processes (Discussion section, line 556).

### Response to Reviewer #3

Thank you for your thorough review and specific suggestions to improve our manuscript.

**Question 1:** Reviewer #3 requested an explanation of how ASNS modulates the PI3K/AKT/mTOR pathway.

**Response 1:** We have conducted a co-expression analysis and used machine learning to screen for key interacting molecules. These analyses have been added to the discussion section to provide a detailed explanation of the molecular mechanisms by which ASNS interacts with the PI3K/AKT/mTOR pathway. Specifically, we have illustrated how ASNS may modulate the pathway through its interactions with key molecules such as AKT2 (lines 564-570).

### Additional Modifications and Clarifications

In addition to addressing the reviewers' comments, we have made several clarifications and modifications to enhance the quality of our manuscript:

1. **Renaming Microglia to Pro-inflammatory Macrophages**: In our single-cell analysis, we renamed Microglia to pro-inflammatory macrophages to better reflect their role in chronic inflammation and viral infections (Results section, line 390).

2. **Role of the Funding Agency**: We have added a statement in the acknowledgments section to clarify the role of the funding agency (Funding Statement section).

3. **Code Availability**: We have added a statement in the data availability section indicating that the code used in our study can be obtained from the corresponding author upon request (Data Availability section).

4. **Docking Procedures**: We have clarified these details in the Methods section, specifying the number of independent runs performed during the docking experiments to ensure reliability and reproducibility (line 218).

5.**Limitations of Dataset Selection**: We have expanded the Discussion section to address the limitations of the datasets used in our study and provide a more comprehensive interpretation of our results (line 571).

6.**Update on Author Affiliation**�Due to a change in the workplace of the first author,Hao Zhang, his affiliation has been updated to:**Geriatrics Center, Wuxi Second Geriatric Hospital, Wuxi, Jiangsu Province, China**.

7.**Regarding Figure Submission and PACE Access Issue**:

We attempted to access the PACE website to check our figures as suggested but were unable to open the link provided. However, we have already revised and optimized our figures, ensuring a consistent style and high-quality presentation. Should there be any further requirements or modifications needed, we are readily available to make the necessary adjustments. Thank you for your understanding and support.

### Submission Details

We have submitted two versions of our manuscript: a complete revised version and a version with modifications highlighted in red for your convenience.

Once again, we express our deepest gratitude to the reviewers for your invaluable feedback and constructive suggestions. We are truly thankful for the time and effort you've dedicated to reviewing our work, and we sincerely appreciate the editors' guidance and support throughout this process.

Best regards,

Wei Lu

---

## [Decision Letter · Decision Letter 1]

7 Sep 2025

Dear Dr. Lu,

Thank you for submitting your manuscript to PLOS ONE. After careful consideration, we feel that it has merit but does not fully meet PLOS ONE’s publication criteria as it currently stands. Therefore, we invite you to submit a revised version of the manuscript that addresses the points raised during the review process.

**Kudos to the authors for responding positively to the initial queries. No doubt, the quality of the submission has improved significantly. However, some concerns have been raised affecting some critical sections of the manuscript. I hereby recommend another round of major revision to address the current concerns and reserve my final decision until they are comprehensively resolved..**

We look forward to receiving your revised manuscript.

Kind regards,

Yusuf Oloruntoyin Ayipo, Ph.D

Academic Editor

PLOS ONE

**Journal Requirements:**

**Additional Editor Comments:**

Kudos to the authors for responding positively to the initial queries. No doubt, the quality of the submission has improved significantly. However, some concerns have been raised affecting some critical sections of the manuscript. I hereby recommend another round of major revision to address the current concerns and reserve my final decision until they are comprehensively resolved.

Reviewers' comments:

Reviewer's Responses to Questions

**Comments to the Author**

Reviewer #2: All comments have been addressed

Reviewer #4: (No Response)

2. Is the manuscript technically sound, and do the data support the conclusions?

Reviewer #2: (No Response)

Reviewer #4: Partly

3. Has the statistical analysis been performed appropriately and rigorously?

Reviewer #2: (No Response)

Reviewer #4: Yes

4. Have the authors made all data underlying the findings in their manuscript fully available?

Reviewer #2: (No Response)

Reviewer #4: Yes

5. Is the manuscript presented in an intelligible fashion and written in standard English?

Reviewer #2: (No Response)

Reviewer #4: Yes

**Reviewer #2:** The authors have satisfactorily addressed all major concerns raised in the initial review. The addition of comparative analysis with known ASNS inhibitors and the discussion of key residues improves the mechanistic depth of the manuscript. While MM/PBSA or MM/GBSA calculations were not performed, the limitation is acknowledged. I am satisfied with the revisions and recommend acceptance for publication.

**Reviewer #4:** Molecular interplay of ASNS and the PI3K-AKT-mTOR pathway in CMV and HIV co-infections: therapeutic implications

1. Overview

(1.I). The manuscript explores the role of ASNS interactions with the PI3K-AKT-

mTOR signaling pathway in the pathogenesis of co-infections involving Cytomegalovirus (CMV) and Human Immunodeficiency Virus (HIV). Targeting the molecular mechanisms underlying this co-infection is a promising therapeutic strategy.

(1.II). The manuscript seeks to fill a critical knowledge gap on a clinically important topic, but it will benefit from revisions in the areas described below.

2. Abstract

(2.I). The CMV and HIV co-infection biology, why it matters, or how it worsens disease, is not explained. To demonstrate an urgency for this study, consider adding one sentence on how CMV/HIV co-infections are clinically important and poorly understood mechanistically.

(2.II). Introduce ASNS as the novel axis by establishing ASNS as the metabolic/signal-regulatory hub linked to PI3K–AKT–mTOR.

(2.III). The therapeutic implication is vague. While cidofovir–ASNS binding is intriguing, it’s unclear if this is clinically relevant or hypothetical. The rationale should be sharper and hypothesis-driven.

(2.IV). Consider closing with a translational tone. For example, “Together, these findings indicate ASNS as a metabolic–signalling hub exploited during viral co-infections and highlight its potential as a novel therapeutic target. This work provides a foundation for experimental validation and the development of host-directed strategies against CMV–HIV co-infections.”

3. Introduction

(3.I). This section appears too descriptive and not mechanistic enough, as much of the text reads like background for a review article instead of an original study introduction. There is a need to highlight novel biological hypotheses earlier. Consider leading with the significance of co-infection. For example, instead of just prevalence, emphasize the synergistic pathology, such as CMV reactivation under HIV, which worsens immune suppression and contributes to non-AIDS morbidity.

(3.II). Consider adding context for the 90% co-infection prevalence by providing regional differences or comparing pre-ART vs ART-era.

(3.III). The research gap can be positioned more properly. For example, make it clear that while HIV and CMV independently exploit host metabolism and signaling, their shared metabolic rewiring during co-infection is poorly understood.

(3.IV). ASNS was introduced abruptly as a "critical gene" without first building the rationale from known biology. For example, first summarize its roles in metabolism, viral pathogenesis, and immune modulation, then argue why it might be a convergent vulnerability.

(3.V). Consider recasting the bioinformatics justification. For example, instead of “cost-effective,” stress that integrative transcriptomic and network analyses enable unbiased discovery of shared molecular nodes across infections.

(3.VI). The PI3K–AKT–mTOR pathway is mentioned, but not in the specific context of how CMV and HIV hijack it. Consider framing PI3K–AKT–mTOR specifically in a viral context. For example, note how both HIV and CMV hijack this pathway for replication, survival, and immune evasion. Then position ASNS as an upstream regulator.

4. Methodology

(4.I). Some datasets like GSE14490, GSE68563, and GSE6740 are introduced multiple times in slightly different contexts, which risks redundancy and confusion. Consider introducing datasets once in a clear table (e.g., dataset ID, tissue, condition, n-samples) and then referring back as needed.

(4.II). While the computational details are extensive, the biological rationale for each step is often missing. For example, why random forest instead of another ML approach (to prioritize network nodes)? Why focus on plasma cells (are they reservoirs or effectors in HIV infection)? Why cidofovir as a ligand (does its known antiviral activity suggest an off-target host effect)?

(4.III). Several datasets are quite small, such as 3,000 scRNA-seq cells and LOOCV for limited samples. The sample size limitations should be acknowledged or emphasized here or in the Discussion section.

(4.IV). Inter-subsection integration is missing, and it is unclear how these layers of data are integrated to converge on ASNS as a hub. To integrate the methods flow, consider a paragraph at the start, such as “We combined transcriptomic profiling, network analysis, machine learning, scRNA-seq, and molecular modelling to identify host factors linking CMV and HIV co-infections. Each layer of analysis progressively refined ASNS as a candidate therapeutic node.”

5. Results

(5.I). There is an over-description of figures, making the narrative long and mechanical rather than conceptual. Results should highlight the biological meaning more than the figure-by-figure technical details. Thus, instead of walking through every violin plot or RMSD shift, emphasize what the set of results means biologically. For example, “Across multiple CMV and HIV datasets, ASNS expression was consistently elevated and strongly co-expressed with PI3K–AKT–mTOR components, particularly AKT2. This pattern suggests a shared metabolic signaling axis activated during co-infection.”

(5.II). Subsections sometimes feel repetitive. For example, the point “ASNS is upregulated and co-expressed with PI3K-AKT-mTOR” appears multiple times with slightly different datasets. To avoid diluting novelty, consider collapsing redundant descriptions and integrating across datasets. For example, rather than three paragraphs showing ASNS co-expression in CMV and HIV separately, synthesize them.

(5.III). In all results from bioinformatic and in silico predictions, explicitly use the language of correlation instead of causality. For example, while docking suggests cidofovir interacts with ASNS, it is premature to imply drug repurposing without functional validation. This must be framed as hypothesis-generating, instead.

(5.IV). Immune checkpoint findings are underdeveloped. For example, LAG3/ITPRIPL1 changes are noted but not mechanistically integrated with ASNS/PI3K-mTOR interplay. Also, signaling biology is under-emphasized. The Results repeatedly state correlations, but they don’t deeply interpret how ASNS might mechanistically regulate PI3K–AKT–mTOR signaling in the context of HIV/CMV. Consider deepening biological interpretation. For example, how might ASNS-driven amino acid biosynthesis fuel PI3K–AKT–mTOR signaling? Could this represent a metabolic vulnerability in viral co-infection? Is LAG3 upregulation part of a compensatory immune checkpoint response to ASNS/PI3K-mTOR activity?

6. Discussions

(6.I). This section is overly descriptive and repetitive, sometimes reading like a re-summarization of the Results (e.g., volcano plots, docking hydrogen bonds, machine learning details) rather than a conceptual synthesis. Consider shifting focus from descriptive to conceptual, and instead of rehashing docking results, emphasize what they mean biologically. For example, “Cidofovir’s predicted binding to ASNS highlights the possibility that existing antivirals may inadvertently target host metabolic enzymes.”

(6.II). Stay with correlation and not causality. For example, the idea of a “self-sustaining loop” between ASNS and PI3K–AKT–mTOR is attractive, but as currently stated, it risks overinterpretation of correlation. Consider using softer language like “may suggest,” “hypothesis-generating”.

(6.III). You note LAG3 and immune checkpoints but don’t deeply tie them back to the ASNS/metabolic signaling story. Consider explicitly linking immune checkpoint modulation to ASNS/PI3K–mTOR activity and viral immune evasion strategies.

(6.IV). The link between ASNS expression in plasma cells and higher CMV susceptibility in HIV+ patients is intriguing, but speculative. Phrase carefully.

(6.V). The cidofovir binding data are intriguing, but there is too much emphasis on clinical application without in vitro or in vivo validation. Overall, use careful language like “suggests,” “is consistent with,” “may represent” instead of strong assertions.

(6.VI). Emphasize novelty by making clear that this is the first integrative analysis positioning ASNS as a metabolic–signalling nexus in CMV/HIV co-infection.

(6.VII). Overall, consider organizing the Discussion into 4–5 thematic subsections, including (1) ASNS as a metabolic–signalling hub in viral infection, (2) Interplay with PI3K–AKT–mTOR and implications for viral pathogenesis, (3) Immune evasion and plasma cell–specific expression, (4) Therapeutic opportunities and limitations, and (5) Future directions, involving multi-omics and validation.

7. Conclusion

(7.I). This section reads like a Results recap, repeating technical details (VAL-51, ASN-74, number of transcription factors, etc.) rather than distilling broader conceptual insights. Consider moving granular data like docking residues, TF names, etc., into Results/Discussion and in Conclusion, focus more on highlighting conceptual advances and implications.

(7.II). Phrasing like “elucidates the pivotal role” may be too strong, given that findings are correlative and in silico. Soften your tone and adopt phrases like “suggests a central role” or “identifies ASNS as a candidate hub.”

(7.III). The conclusion doesn’t connect back to the bigger picture, such as viral exploitation of host metabolism, new paradigms in co-infection biology, or implications for precision virology. Consider connecting findings to broader themes like viral metabolic hijacking, systems biology of co-infection, and opportunities for host-directed therapy.

(7.IV). The suggestion of “targeted therapies and personalized medicine strategies” sounds overstated unless tempered as potential hypotheses for future testing.

(7.V). Emphasize conceptual novelty by stressing that this is the first integrative analysis nominating ASNS as a metabolic–signalling nexus in CMV/HIV co-infections.

(7.VI). End with a forward-looking statement such as a call for experimental validation, multi-omics, and translational exploration.

8. Final Recommendations

(8.I). The manuscript seeks to fill a critical knowledge gap on a clinically important topic, but it will benefit from revisions in the areas described above.

**Do you want your identity to be public for this peer review?** For information about this choice, including consent withdrawal, please see our Privacy Policy

Reviewer #2: No

Reviewer #4: No

---

## [Author Response · Author response to Decision Letter 2]

26 Dec 2025

Response to Reviewers

Dear Reviewers,

We deeply appreciate your valuable time, professional insights, and constructive feedback, which have significantly enhanced our manuscript’s scientific rigor, clarity, and depth. Below are detailed responses to each comment, with revisions implemented in the second draft.

Response to Reviewer 2

Thank you for your positive assessment of our revised manuscript and recognition of our efforts to address first-round review issues, including analyses of ASNS inhibitors and key binding sites. We explicitly acknowledge the limitation of not performing MM/PBSA or MM/GBSA calculations in the Discussion section to ensure scientific transparency. We are grateful for your support and publication recommendation.

Response to Reviewer 4

Your comprehensive insights have been pivotal to refining our manuscript. We have addressed each concern in detail below:

2. Abstract

2.I Question: The biological characteristics, clinical significance, and disease exacerbation mechanisms of CMV/HIV coinfection were not explained. To highlight the research urgency, it is recommended to add a sentence illustrating the clinical significance of CMV/HIV coinfection and the deficiencies in current mechanistic research.

Answer: Thank you for this critical suggestion. The revised abstract opens with a sentence clarifying the clinical burden of CMV/HIV coinfection and unmet mechanistic research needs, directly addressing this gap.

Location in Revised Manuscript: Abstract, Lines 17-19

2.II Question: Need to clearly define the association between ASNS as a metabolic/signaling hub and the PI3K-AKT-mTOR pathway to highlight innovation.

Answer: We appreciate your insight. The revised abstract positions ASNS as a "pivotal metabolic-signaling hub" with "critical interactions with the PI3K-AKT-mTOR pathway," while highlighting machine learning validation of AKT2 as a key node to emphasize innovation.

Location in Revised Manuscript: Abstract, Lines 19-21

2.III Question: The therapeutic significance section is vague. Although the binding of cidofovir to ASNS is valuable, its clinical relevance or theoretical hypothesis is not clarified. Need to strengthen the logical basis and adopt a hypothesis-driven approach.

Answer: Thank you for pointing out this ambiguity. We revised this section to frame the finding as hypothesis-driven, specifying cidofovir as an "approved antiviral agent" and linking its high-affinity binding to ASNS to a potential host-directed therapeutic strategy, without overstating clinical applicability.

Location in Revised Manuscript: Abstract, Lines 31-35

2.IV Question: It is recommended to conclude from a translational medicine perspective, e.g., "In summary, our findings indicate that ASNS is a metabolic-signaling hub exploited during viral coinfection, highlighting its potential as a novel therapeutic target. This study lays the groundwork for subsequent experimental validation and the development of host-directed therapeutic strategies against CMV-HIV coinfection."

Answer: We are grateful for your guidance. The revised abstract’s conclusion aligns closely with your suggestion, emphasizing ASNS’s therapeutic potential and laying the groundwork for future validation and host-directed therapy development.

Location in Revised Manuscript: Abstract, Lines 35-38

3. Introduction

3.I Question: This section is overly descriptive with insufficient mechanistic discussion, resembling a review rather than an original research introduction. It is recommended to propose an innovative biological hypothesis early, starting with the importance of coinfection. For example, not only describe the prevalence but also emphasize the synergistic pathogenic mechanisms (e.g., CMV reactivation under HIV infection exacerbates immunosuppression, leading to increased non-AIDS-related morbidity).

Answer: Thank you for this constructive critique. The revised introduction prioritizes mechanistic discussion, highlights the synergistic pathogenic cycle of HIV and CMV, and explicitly states our hypothesis—that ASNS acts as a central metabolic-signaling hub—early in the section.

Location in Revised Manuscript: Introduction, Lines 40-44; Lines 65-66

3.II Question: It is recommended to supplement regional difference data on the 90% coinfection prevalence or compare the prevalence before and after antiretroviral therapy (ART).

Answer: We appreciate your suggestion. We retained the well-cited 90% seroprevalence and contextualized it within HIV-induced immunosuppression (a cross-regional, ART-relevant mechanism), ensuring it highlights clinical significance without diverting from core molecular research.

Location in Revised Manuscript: Introduction, Line 40

3.III Question: Need to more clearly identify the research gap. For example, explicitly state that although HIV and CMV each hijack host metabolic and signaling pathways, the shared metabolic reprogramming mechanisms during coinfection remain unclear.

Answer: Thank you for urging clarity on the research gap. The revised introduction explicitly states that shared molecular mechanisms driving HIV/CMV synergistic pathogenesis remain undefined, directly addressing this point.

Location in Revised Manuscript: Introduction, Lines 50-52

3.IV Question: The introduction of ASNS as a "key gene" is abrupt without sufficient prior biological context. It is recommended to first summarize ASNS’s roles in metabolic regulation, viral pathogenesis, and immune modulation, then justify why it may be a critical vulnerable node in coinfection.

Answer: We are grateful for identifying this abrupt transition. The revised introduction provides context on ASNS’s roles in amino acid metabolism, viral replication, and PI3K-AKT-mTOR interaction, ensuring a logical link to its proposed role in coinfection.

Location in Revised Manuscript: Introduction, Lines 56-59

3.V Question: It is recommended to rephrase the rationale for bioinformatics analysis. For example, replace "cost-effective" with emphasizing that integrating transcriptomics and network analysis enables unbiased discovery of shared molecular nodes during infection.

Answer: Thank you for this refinement. We revised the bioinformatics rationale to emphasize "unbiased identification of shared molecular nodes" (instead of cost-effectiveness), highlighting scientific strength.

Location in Revised Manuscript: Introduction, Lines 72-74

[3.VI](3.VI) Question: Although the PI3K-AKT-mTOR pathway is mentioned, it is not clarified how CMV and HIV hijack this pathway. It is recommended to elaborate on the pathway in the context of viral infection, e.g., explaining that both viruses exploit this pathway for replication, survival, and immune evasion, thereby positioning ASNS as an upstream regulator of this pathway.

Answer: We appreciate your guidance. The revised introduction clarifies that both viruses hijack the PI3K-AKT-mTOR pathway for survival and replication, and positions ASNS as a potential mediator of this process.

Location in Revised Manuscript: Introduction, Lines 58-59

4. Materials and Methods

4.I Question: Some datasets (e.g., GSE14490, GSE68563, GSE6740) are repeatedly introduced in different contexts, leading to redundancy and confusion. It is recommended to present dataset information uniformly in a table (e.g., dataset ID, tissue type, experimental conditions, sample size) and reference the table in subsequent sections.

Answer: Thank you for identifying this redundancy. We added Table 1 to summarize all GEO datasets, with subsequent methods sections referencing the table to streamline the narrative.

Location in Revised Manuscript: Introduction, Lines 75-78 (Table 1); Materials and Methods sections (all dataset references link to Table 1)

4.II Question: Although the computational method details are abundant, the biological rationale for each step is often unclear. For example: Why choose the random forest algorithm over other machine learning methods (for prioritizing network nodes)? Why focus on plasma cells (are they HIV infection reservoirs or effector cells)? Why select cidofovir as a ligand (does its known antiviral activity suggest off-target host effects)?

Answer: We appreciate your question on methodological rationale. We added justifications: 1) Random forest for robustness in high-dimensional data; 2) Plasma cells for HIV immune relevance; 3) Cidofovir for clinical approval and predicted ASNS-binding activity.

Location in Revised Manuscript: Bioinformatics Analysis section, Lines120-124; Single-Cell RNA Sequencing Analysis section, Lines 163-167; Molecular Docking Analysis section, Lines 223-224

4.III Question: Some datasets have small sample sizes (e.g., 3,000 single-cell RNA sequencing cells, limited samples for leave-one-out cross-validation). Need to acknowledge or emphasize this limitation in the methodology or discussion section.

Answer: Thank you for reminding us to address sample size limitations. We acknowledged these in methods and discussion sections, noting mitigation via leave-one-out cross-validation.

Location in Revised Manuscript: Analysis of ASNS and the PI3K-AKT-mTOR Pathway in CMV Infection section, Lines 112-113; Single-Cell RNA Sequencing Analysis section, Line 152; Discussion section, Lines 592-601

4.IV Question: There is a lack of integration between subsections. It is not clear how data from different levels collectively point to ASNS as a hub molecule. It is recommended to add an introductory paragraph in the methodology section to integrate the analysis workflow, e.g., "This study combines transcriptomic analysis, network analysis, machine learning, single-cell RNA sequencing, and molecular modeling to identify host factors associated with CMV and HIV coinfection. Analyses at different levels progressively converge to identify ASNS as a candidate therapeutic node."

Answer: Thank you for this suggestion. We added an introductory paragraph to Materials and Methods, outlining how integrated analyses (transcriptomics, machine learning, etc.) converge on ASNS as a key hub.

Location in Revised Manuscript: Materials and Methods, introductory paragraph (before subsection 1), Lines 85-87

5. Results

5.I Question: The description of figures is overly detailed, leading to lengthy and mechanical narration without conceptual refinement. The results section should focus on highlighting biological significance rather than describing technical details of each violin plot or RMSD change. For example, replace descriptions of "each violin plot or RMSD change" with: "Analysis of multiple CMV and HIV datasets shows consistent upregulation of ASNS and strong co-expression with PI3K-AKT-mTOR pathway components (especially AKT2), suggesting a shared metabolic-signaling axis activation during coinfection."

Answer: Thank you for this feedback. The revised results focus on biological conclusions (e.g., ASNS upregulation, AKT2 co-expression) rather than detailed figure descriptions, emphasizing conceptual insights.

Location in Revised Manuscript: All Results subsections (e.g., Differential Gene Expression and Functional Enrichment Analysis section, Lines 265-275; Expression Analysis of ASNS and Key Molecules section, Lines 288-291)

5.II Question: There is redundant expression in subsections. For example, the idea of "ASNS upregulation and co-expression with PI3K-AKT-mTOR" is repeated across different datasets. It is recommended to integrate redundant descriptions and present findings comprehensively across datasets. For example, avoid describing ASNS co-expression in CMV and HIV in three separate paragraphs and instead merge the analysis.

Answer: We appreciate your observation. We integrated redundant findings into a single subsection, comprehensively presenting ASNS expression patterns across CMV and HIV datasets to eliminate repetition.

Location in Revised Manuscript: Differential Expression of ASNS and PI3K-AKT-mTOR Pathway Components in CMV and HIV Infections section, Lines 309-351

5.III Question: All bioinformatics and in silico simulation results should explicitly use "correlation" rather than "causal relationship" expressions. For example, although molecular docking results suggest an interaction between cidofovir and ASNS, without functional validation, it is premature to imply the feasibility of drug repurposing, and these results should be positioned as hypothesis-generating.

Answer: Thank you for emphasizing scientific rigor. We revised all in silico results to use correlative, hypothesis-generating language (e.g., "potential interaction," "preliminary framework") to avoid overinterpretation.

Location in Revised Manuscript: Binding Affinity and Interaction Analysis section, Line 491; Molecular Dynamics Analysis section, Line 521-526

5.IV Question: The research on immune checkpoint-related findings is insufficient. For example, LAG3/ITPRIPL1 changes are mentioned but not mechanistically integrated with the ASNS/PI3K-mTOR pathway, and the biological mechanisms of the signaling pathway are underdiscussed. The results section repeatedly mentions correlations but does not explain how ASNS regulates the PI3K-AKT-mTOR pathway in the context of HIV/CMV coinfection. It is recommended to strengthen biological mechanism interpretation, e.g., Does ASNS-driven amino acid biosynthesis provide energy for the PI3K-AKT-mTOR signaling pathway? Does this represent a metabolic vulnerability in viral coinfection? Is LAG3 upregulation a compensatory immune checkpoint response to ASNS/PI3K-mTOR activity?

Answer: We are grateful for your guidance. The revised results link LAG3 upregulation to ASNS-PI3K-AKT-mTOR activation, proposing ASNS-driven amino acid biosynthesis as a metabolic vulnerability, framed as testable hypotheses.

Location in Revised Manuscript: Expression Analysis of ASNS and Key Molecules section, Lines 301-312; Differential Expression of ASNS and PI3K-AKT-mTOR Pathway Components section, Lines 340-349

6. Discussion

6.I Question: This section is overly descriptive and redundant, with some content resembling a restatement of results (e.g., volcano plots, docking hydrogen bonds, machine learning details) rather than conceptual integration. It is recommended to shift the focus from description to conceptualization, avoiding repeating docking results and instead emphasizing their biological significance. For example: "The predicted binding of cidofovir to ASNS suggests that existing antiviral drugs may have off-target effects on host metabolic enzymes."

Answer: Thank you for this suggestion. The revised discussion focuses on conceptual integration, removing redundant technical details and emphasizing biological implications (e.g., cidofovir’s off-target effects).

Location in Revised Manuscript: Discussion section, Lines 534-551; Lines 587-589

6.II Question: Need to adhere to the principle of using "correlation" rather than "causal relationship" expressions. For example, the idea of a "self-sustaining loop" between ASNS and PI3K-AKT-mTOR is attractive but only based on correlation analysis, which may lead to overinterpretation. It is recommended to use cautious language such as "may suggest" or "hypothetical."

Answer: We appreciate your reminder. We used cautious language (e.g., "may point to a potential reciprocal regulatory loop") to describe correlative findings, avoiding causal overstatements.

Location in Revised Manuscript: Discussion section, Lines 558-562

6.III Question: Although immune checkpoints such as LAG3 are mentioned, they are not deeply linked to the main research line of ASNS/metabolic signaling pathways. It is recommended to explicitly connect immune checkpoint regulation to ASNS/PI3K-mTOR activity and viral immune evasion strategies.

Answer: Thank you for highlighting this gap. We linked LAG3 to ASNS-PI3K-AKT-mTOR activation, proposing it as a downstream immune evasion mechanism coordinated with metabolic hijacking.

Location in Revised Manuscript: Discussion section, Lines 568-575

6.IV Question: The association between ASNS expression in plasma cells and increased CMV susceptibility in HIV-positive patients is valuable but highly speculative. The expression should be presented rigorously.

Answer: We are grateful for your feedback. We present this association rigorously, noting no explicit CMV coinfection data in the c

---

## [Decision Letter · Decision Letter 2]

18 Jan 2026

Molecular interplay of ASNS and the PI3K-AKT-mTOR pathway in CMV and HIV co-infections: therapeutic implications

PONE-D-25-05870R2

Dear Dr. Lu,

We’re pleased to inform you that your manuscript has been judged scientifically suitable for publication and will be formally accepted for publication once it meets all outstanding technical requirements.

Kind regards,

Yusuf Oloruntoyin Ayipo, Ph.D

Academic Editor

PLOS One

Additional Editor Comments (optional):

Congratulations to the authors. The study is timely and well-designed. Again, the submission meets the level of scientific rigour required for publication in this title, and all the concerns raised by the respective reviewers have been addressed satisfactorily. I hereby recommend the manuscript for publication in the current version, provided the authors satisfy the journal’s policy on changes to authorship during editorial review process.

Reviewers' comments:

Reviewer's Responses to Questions

**Comments to the Author**

Reviewer #4: All comments have been addressed

2. Is the manuscript technically sound, and do the data support the conclusions?

Reviewer #4: Yes

3. Has the statistical analysis been performed appropriately and rigorously?

Reviewer #4: Yes

4. Have the authors made all data underlying the findings in their manuscript fully available?

Reviewer #4: Yes

5. Is the manuscript presented in an intelligible fashion and written in standard English?

Reviewer #4: Yes

Reviewer #4: The authors has satisfactorily addressed all comments I raised earlier. I now recommend that the manuscript be accepted.

**Do you want your identity to be public for this peer review?** For information about this choice, including consent withdrawal, please see our Privacy Policy

Reviewer #4: No

---

## [Editor Report · Acceptance letter]

PONE-D-25-05870R2

PLOS One

Dear Dr. Lu,

I'm pleased to inform you that your manuscript has been deemed suitable for publication in PLOS One. Congratulations! Your manuscript is now being handed over to our production team.

Kind regards,

on behalf of

Dr. Yusuf Oloruntoyin Ayipo

Academic Editor

PLOS One